

# Interpretable Soil Moisture Prediction with a Physics-guided Deep Learning Approach

Yanling Wang[1], Xiaolong Hu[1*], Yaan Hu[2], Leilei He[1], Lijun Wang[1], Wenxiang Song[1], Liangsheng Shi[1*]

[1]State Key Laboratory of Water Resources Engineering and Management, Wuhan University, Wuhan, China

[2]State Key Laboratory of Hydrology-Water Resources and Hydraulic Engineering, Nanjing Hydraulic Research Institute, Nanjing, China

*Corresponding author*: Xiaolong. Hu (xlhu@whu.edu.cn), Liangsheng Shi (liangshs@whu.edu.cn)

**Abstract.** Soil moisture is a critical component of the hydrological cycle, but accurately predicting it remains challenging due to the nonlinearity of soil water transport, variability in boundary conditions, and the intricate nature of soil properties. Recently, deep learning has shown promise in this domain, typically by modeling temporal dependencies for soil moisture predictions. In this study, we propose non-local neural networks (NLNN) to convert this problem into a single-time-step, simultaneous multi-

depth soil moisture forecasting. By facilitating mutual compensation among different depths, this method enables a representation of vertical heterogeneity and inter-layer connectivity without physical assumptions, leading to precise and efficient predictions in diverse scenarios. Our non-local operation design includes the embedded Gaussian operations and disentangled physics-guided operations, resulting in two variants: the self-attention non-local neural network (SA-NLNN) and the physics-guided non-

local neural network (PG-NLNN). The models offer visual interpretability, providing insights into intricate mechanisms of soil moisture dynamics. Notably, the model guided by physics yields more stable and reasonable qualitative interpretations. With in-situ observations, we demonstrate that our proposed models perform satisfactorily. The physics-guided non-local operations significantly enhance accuracy and reliability. Additionally, our models adapt to diverse time-scale situations while maintaining high computational efficiency. Both models exhibit robust noise resistance, with physics guidance enhancing

PG-NLNN's noise resistance. In summary, our work addresses the soil moisture prediction challenge in a novel way, highlighting the potential of NLNN and the importance of incorporating physic guidance in data-driven models.



*Keywords:* soil moisture; deep learning; non-local neural networks; physics-guided; visual interpretability

### 1. Introduction

Soil moisture plays an important role in hydrological processes, governing the exchange of water and energy fluxes between the atmosphere and the land (Vereecken et al., 2008). Accurate simulations of soil moisture dynamics hold great significance in various domains, including effective water resources planning and management, agricultural production, and flood disaster monitoring (Entekhabi et al., 1996; Koster et al., 2004; Zhang et al., 2018). However, precisely forecasting soil moisture dynamics poses challenges due to the nonlinearity of soil water transport (Richards, 1931), randomness in boundary conditions (Guswa et al., 2002), and the intricate nature of soil properties, including soil structure and hydraulic parameters (Vereecken et al., 2022). These factors contribute to strong spatio-temporal variabilities in soil moisture dynamics (Heathman et al., 2012). Traditionally, the simulation of soil moisture dynamics has primarily relied on physically based models, such as the soil-plant-atmosphere-water model (Saxton et al., 1974) and HYDRUS (Simunek et al., 2005). However, their implementation faces challenges in accurately estimating the required parameters (Bandai & Ghezzehei, 2021; Gill et al., 2006). What's more, the current methodology struggles to accurately characterize soil structure at spatially relevant scales (Romero-Ruiz et al., 2018). This limitation complicates handling scenarios involving cracks, root water absorption, and other complexities, as illustrated in Figure 1. With advancements in technology and big data analysis capabilities, data-driven models have aroused increasing focus and appear to be more practical in soil moisture dynamics forecasting. For instance, researchers have discovered that both support vector regression and random forest show satisfactory results in soil moisture prediction while maintaining low computing costs (Gill et al., 2006; Prasad et al., 2019). Furthermore, the extreme learning machine (Huang et al., 2006) has demonstrated its capability to precisely predict soil moisture trends (Y. Liu et al., 2014).

In recent years, deep learning (Lecun et al., 2015) has gained considerable attention for its remarkable capabilities in fitting to complex data patterns. When predicting soil moisture, deep learning primarily relies on modeling temporal dependencies. The fundamental models handling sequential data fall into





three categories: Recurrent Neural Networks (RNNs) (Elman, 1990), Convolution Neural Networks (CNNs) (LeCun, 1989), and Transformers (Vaswani et al., 2017). RNNs exploit temporal dependencies through recurrent operations, with Long Short-Term Memory (LSTM) networks demonstrating accurate

soil moisture predictions (Fang et al., 2019). CNNs capture dependencies with repetitive convolutional operations and also yield satisfactory results in soil moisture dynamics modeling (Severyn & Moschitti, 2015; Shi et al., 2015). Both recurrent and convolutional operations process local neighborhoods in input data. Consequently, long-range dependencies are captured through repeated local operations, which is inefficient (L. Zhu et al., 2021). In contrast, Transformers process data in a more efficient way, owing to

its core component – self-attention mechanisms. These mechanisms extract crucial long-range non-local information directly. For instance, Temporal Fusion Transformers with interpretable self-attention layers have shown significant improvements over existing benchmarks in multi-horizon time series forecasting (Lim et al., 2021). Furthermore, Transformers exhibit potential for effective soil moisture dynamics prediction with straightforward model structures (Y. Wang, Shi, Hu, Hu, et al., 2023). Researchers are

increasingly recognizing the potential of Transformers.

However, it is worth noting that current deep learning models often lack physical laws and interpretability. To bridge the gap between data-driven approaches and physics, physical laws can be embedded into model architectures or loss functions. For instance, Jiang et al. (2020) integrated the physical processes from a conceptual hydrological model into an RNN for runoff modeling. De Bézenac

et al. (2018) incorporated advection-diffusion principles into the kernel design of a CNN to predict sea surface temperature. Additionally, some researchers have added the residuals of the physical governing equations into the loss function, resulting in a novel approach known as Physics-informed Neural Networks (PINN). (M. Raissi et al., 2019; Maziar Raissi et al., 2017). This approach has been applied to soil moisture modeling (Bandai & Ghezzehei, 2021; Y. Wang, Shi, Hu, Song, et al., 2023). Although this

integration enhances model credibility, developing appropriate training strategies to improve extrapolation accuracy remains challenging (Lu et al., 2021). Besides, the complex coupling of actual physical processes and the presence of unknown governing equations pose substantial challenges in practical applications. To date, most previous works have relied on traditional model structures, leaving a critical gap in reliable, physics-guided data-driven methods for soil moisture prediction. This



underscores the necessity of transitioning toward soil science-informed machine learning models that use

the power of data-driven techniques while integrating soil science knowledge during the training process

to enhance reliability and generalizability (Minasny et al., 2024).

Considering that physical models calculate soil moisture content by iteratively using current soil

profile states for step-by-step predictions, we incorporate the spatial interactions of soil moisture within

the profile into our machine learning model, instead of modeling sequential dependencies. Temporal

variations in surface soil moisture exhibit greater variability due to meteorological forcing, while subsoil

moisture dynamics are influenced by soil water redistribution processes (Rosenbaum et al., 2012). When

dealing with relationships between multiple variables, geometric deep learning (Bronstein et al., 2017)

defines model invariances to enhance robustness and generalization. As an example, graph neural

networks (Scarselli et al., 2008) utilize the adjacency matrix to aggregate node features and achieve local

invariance. Inspired by this, we introduce a novel model -- the Non-local Neural Networks (NLNNs) (X.

Wang et al., 2018) to capture spatially invariant soil moisture relationships across depths, thereby

modeling vertical heterogeneity and inter-layer connectivity without physical assumptions. Essentially,

the non-local operation in NLNNs calculates responses at specific locations by aggregating features from

all positions in the input feature map (X. Wang et al., 2018). This allows NLNNs to capture global

dependencies directly and efficiently. According to the definition, NLNNs are flexible and easily

customizable to suit specific requirements. Moreover, the weights computed through non-local

operations provide qualitative interpretation for model learning mechanisms. NLNNs find wide

application in image segmentation tasks and time series forecasting (P. Liu et al., 2019; Z. Zhu et al.,

2019). As a representative of NLNNs, the Transformer is adept at processing various types of data,

including images and video-related challenges (Guo et al., 2022; Khan et al., 2022; Lim et al., 2021; Z.

Liu et al., 2021; Xie et al., 2021). Furthermore, NLNNs can serve as auxiliary blocks to enhance context

modeling abilities (X. Wang et al., 2018; Yin et al., 2020). With the flexibility of non-local operation

modifications, we can envision using NLNNs to simulate the characteristics of soil water dynamics in

spatial distribution while ensuring interpretability.

In this study, we have integrated NLNNs to simulate in-profile soil moisture interactions and predict

multi-depth soil moisture content without physical assumptions. Our aim is to achieve accurate and





effective forecasts under diverse real-world scenarios, as depicted in Figure 1, while also providing interpretability of intricate soil moisture dynamics, such as vertical heterogeneity and inter-layer

connectivity. Specifically, we discard all assumptions on soil, root, or boundary conditions and instead attempt to learn the soil water dynamics directly from the data. Unlike traditional one-dimensional soil water flow models, our model considers real soil moisture interactions across various depths that happen in a three-dimensional soil column, enhancing predictions in complex scenarios. We introduce the Self-Attention mechanism Non-local Neural Networks (SA-NLNN) to explore the potential of NLNN

structures in soil moisture forecasting. Moreover, the Physics-Guided Non-local Neural Network (PG-NLNN) that incorporates soil water transport guidance into the non-local operation is proposed. We examine the models' interpretability using the synthetic data, while in-situ data is applied to assess the practicality and accuracy of the models. The key innovations of our study are as follows: First, unlike previous machine learning models that often focus on time series data processing, our study considers

soil moisture interactions within the profile, converting it into a single-time-step problem involving multi-depth variables. This approach offers mutual compensation within the soil profile, enabling effective and precise soil moisture forecasts. The adaptability of NLNNs across various temporal and spatial scales is also demonstrated. Second, our NLNN models provide interpretable visualizations of non-local weights, offering qualitative descriptions of intricate soil properties derived from the soil

moisture data. The model interpretability is investigated using synthetic soil moisture data, including virtual examples of homogeneous soil, heterogeneous soil, two-layered soil, and soil with root water uptake. Third, incorporating physics-inspired concepts enhances model accuracy and reliability. By integrating meteorological conditions and the spatial interactions of soil moisture within its four-part disentangled physics-guided operation framework, PG-NLNN demonstrates superior performance.

Besides, it offers more reliable interpretations and exhibits robust resistance to noise. When evaluating practical performance, we utilize in-situ soil moisture data sourced from the International Soil Moisture Network (ISMN) and compare our models with the benchmark LSTM model (Datta & Faroughi, 2023; Semwal et al., 2021; Y. Wang, Shi, Hu, Hu, et al., 2023). To the best of our knowledge, this marks the first instance of employing NLNNs for interpretable soil moisture dynamics forecasting.




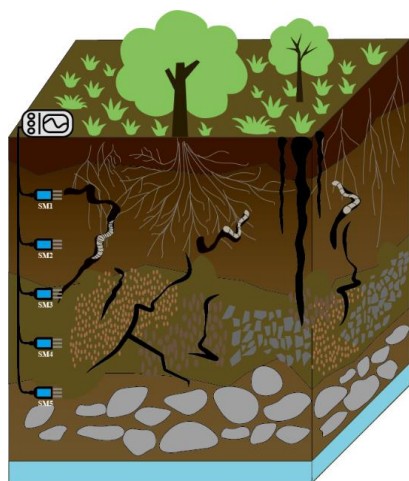

**Figure 1.** Examples of complex soil conditions related to soil texture and soil structure at the soil profile scale. SM3 is more related to SM1 other than SM2 or SM4, due to the existence of wormholes. The proposed non-local neural network is designed to understand that SM3 is highly correlated with SM1 (caused by fast water migration in
wormholes) and less correlated with SM2 (caused by slow seepage under gravity).

In the remainder of this manuscript, Section 2 presents the NLNNs for soil moisture forecasting, including the SA-NLNN and PG-NLNN; Section 3 describes the synthetically generated soil moisture data and the in-situ data; Section 4 provides the model results and the interpretability analysis. Finally,
the conclusion is drawn in Section 5.

## 2. Methodologies

### 2.1 Model structures

In our soil moisture forecasts at multiple depths, we assume that the soil moisture within the profile at the next time step depends on both the current meteorological conditions and the soil moisture from the
previous time step. The NLNN models are designed to capture the potential interactions of soil moisture at different depths within the vertical profile (Figure 1), thereby making predictions that are closer to reality. Figure 2 illustrates the NLNN structure proposed for soil moisture dynamics prediction. The input data for the NLNN model, denoted as $\boldsymbol{sm}^t = [sm_0^t, sm_1^t, sm_2^t, \dots, sm_{n-1}^t, sm_n^t]^{\mathrm{T}}$ comprises a concatenation of soil moisture data at $n$ depths from the previous time step and the upper boundary
condition factor $sm_0^t$ obtained from meteorological conditions processing through an LSTM.



Within our framework, we employ two types of non-local operations. The first type utilizes Gaussian functions in the non-local operation, and the NLNNs composed of Gaussian functions are referred to as SA-NLNN. In the second model, PG-NLNN, the non-local operation is decoupled based on the soil water transport mechanisms. In the NLNN structure, following the non-local operation and a residual connection, a fully connected neural network is employed to generate predictions for the soil moisture at each corresponding depth. This yields predictions denoted as, $sm^{t+1'} = \left[sm_0^{t+1'}, sm_1^{t+1'}, sm_2^{t+1'}, \dots, sm_{n-1}^{t+1'}, sm_n^{t+1'}\right]^{\mathrm{T}}$. The ground truth is represented as $sm^{t+1} = [sm_1^{t+1}, sm_2^{t+1}, \dots, sm_{n-1}^{t+1}, sm_n^{t+1}]^{\mathrm{T}}$. The model is trained by minimizing the error between predictions and the ground truth,

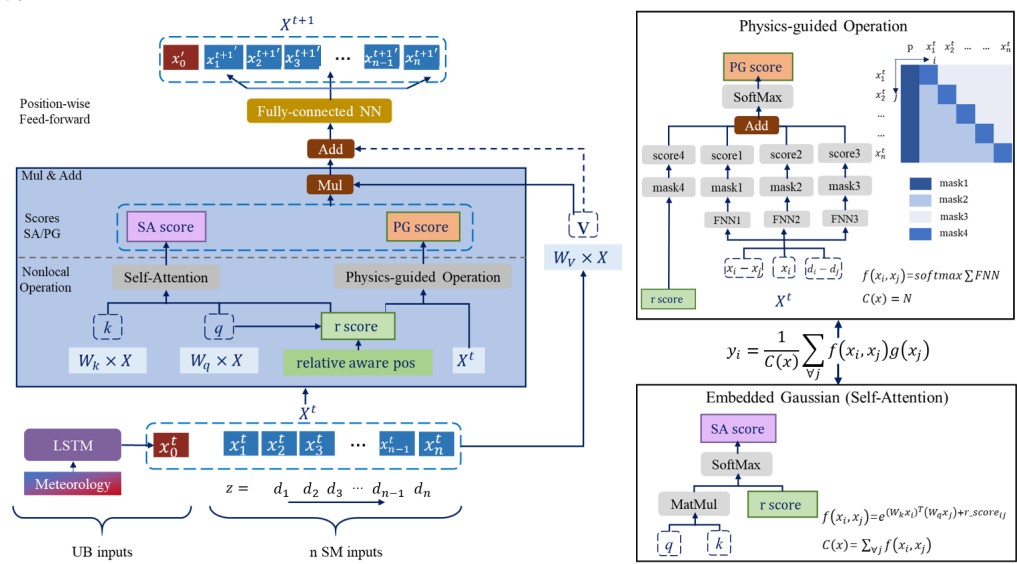

**Figure 2.** Left: non-local neural network structure for soil moisture forecasting. Right: embedded Gaussian operation and physics-guided non-local operation. RPE: relative position encoding. SA/ PG score: non-local weights computed through embedded Gaussian operation and physics-guided operation.

### 2.2 Non-local Operations

The general form of a non-local operation in NLNNs can be defined as follows (X. Wang et al., 2018):

$$y_i = \frac{1}{C(x)} \sum_{\forall j} f(x_i, x_j) g(x_j) \qquad (1)$$



Here $i$ denotes the index of the output $\boldsymbol{y}$ for which the output value is being calculated, while $j$ is the index that lists all conceivable positions in the input $\boldsymbol{x}$. In this context, $\boldsymbol{x}$ represents the input data, and $\boldsymbol{y}$ denotes the corresponding output, sharing the same dimensions as $\boldsymbol{x}$. In this work, $\boldsymbol{x}$ represents the

concatenation of input soil moisture data and the upper boundary condition data, denoted as $\boldsymbol{sm}^t$. $\boldsymbol{x}_i$ and $\boldsymbol{x}_j$ denote the $i_{th}$ and $j_{th}$ indexes in $\boldsymbol{sm}^t$. In other words, $\boldsymbol{x}_i$ and $\boldsymbol{x}_j$ are the soil moisture content at the $i_{th}$ and $j_{th}$ depths, that is $sm_i^t$ and $sm_j^t$. $\boldsymbol{y}$ denotes the output, which corresponds to $\boldsymbol{sm}^{t+1'}$. $\boldsymbol{y}_i$ represents the soil moisture content at $i_{th}$ depth for the next time step $sm_i^{t'}$, which needs to be predicted. The computation of a generic non-local operation involves three components: the pairwise function $f$, the unary function $g$, and the normalization sum $\mathcal{C}(\boldsymbol{x})$. The function $f$ calculates

a scalar (representing relationship such as affinity) between $i$ and all $j$, while the unary function $g$ generates a representation of the input at position $j$. The response is then normalized by $\mathcal{C}(\boldsymbol{x})$. We restrict the form of $g$ to a linear embedding: $g(\boldsymbol{x}_j) = W_g \boldsymbol{x}_j$, where $W_g$ is a weight matrix to be learned. The primary modification focuses on the pairwise function $f$. The $\mathcal{C}(\boldsymbol{x})$ is contingent on the design of $f$.

Following the definition of attention heads from previous work on self-attention mechanisms (Vaswani et al., 2017), our NLNN models employ several operation heads to enhance the model's feature extraction and representation capabilities. The number of operation heads is denoted as $n_{head}$. Similar non-local operations are performed in each head, with some parameter matrices being unique. To form the output, results from each head are concatenated, and a parameterized linear transformation is applied.

It is evident that non-local operations offer flexibility by assuming various forms and can adapt to specific problem designs. This provides potential solutions for many complex situations. In the following sections, we will introduce the classical embedded Gaussian operation, along with our physics-guided non-local operation designed for soil moisture dynamics.

### 2.2.1 Embedded Gaussian Operation:

Self-attention is a specific case of non-local operations within the embedded Gaussian version. It excels in processing data concisely and capturing intricate relationships, making it widely applied in various research areas (Devlin et al., 2019; Lim et al., 2021; Z. Liu et al., 2021). However, it overlooks the ordering of input, necessitating the incorporation of position information into the calculations to ensure accurate processing.



Common position encoding methods include absolute position encoding (Devlin et al., 2019; Gehring et al., 2017; Vaswani et al., 2017) and relative position encoding (Shaw et al., 2018). Absolute position encoding directly incorporates absolute position information pertaining to $i$ or $j$ and integrates it into the input. In contrast, relative position encoding focuses on the relative relationship between position $i$ and $j$. Given the complexity of soil properties and the nature of soil moisture interactions, prioritizing

the relative influence of soil moisture at each depth may prove more effective than relying on absolute position information in soil moisture analysis. In this approach, we utilize the relative position encoding similar to the method proposed by Shaw et al. (2018). The function $f$ encompasses a Gaussian function of two embeddings along with the relative position representation associated with $i$ and $j$. A self-attention mechanism with relative position encodings in each head can be defined as follows:

$$f(\boldsymbol{x}_i, \boldsymbol{x}_j) = e^{((W_k \boldsymbol{x}_j)^{\mathrm{T}}(W_q \boldsymbol{x}_i) + r\_score_{ij})/\sqrt{d_k}} \tag{2}$$

$$\mathcal{C}(\boldsymbol{x}) = \sum_{\forall j} f(\boldsymbol{x}_i, \boldsymbol{x}_j) \tag{3}$$

Here, $W_q$ and $W_k$ are the weight matrixes to be learned for embeddings. $\sqrt{d_k}$ denotes the scale factor, where $d_k$ represents the dimension of the embeddings. $r\_score_{ij}$ is the relative position score computed using relative position encoding. Then the $\boldsymbol{y}_i$ can be calculated through Equation (1). The embedded Gaussian operation for soil moisture forecasts is illustrated in Figure 2.

In the relative position encoding, each relationship between two arbitrary positions $i$ and $j$ is

represented by a learnable vector. Then, the $r\_score_{ij}$ is calculated as follows:

$$r\_score_{ij} = (\boldsymbol{a}_{i,j})^T (W_q \boldsymbol{x}_i) \tag{4}$$

where $\boldsymbol{a}_{i,j}$ represents the relative position encoding utilized for $r\_score_{ij}$ computing. $\boldsymbol{a}_{i,j}$ is a parameter vector that needs to be trained. In the proposed SA-NLNN model, our trainable relative position encoding matrix $A$ consists of $(n+1) \times (n+1)$ distinct elements. The matrix $A$ needs to be learned through training:

$$A = \begin{pmatrix} \boldsymbol{a}_{0,0} & \cdots & \boldsymbol{a}_{0,n} \\ \vdots & \ddots & \vdots \\ \boldsymbol{a}_{n,0} & \cdots & \boldsymbol{a}_{n,n} \end{pmatrix} \tag{5}$$

In this model, all operation heads perform similar operations. $W_q$, $W_k$, and $W_g$ are unique in each head. However, the relative position encoding can be shared across non-local operation heads.



### 2.2.2 Disentangled Physics-Guided operation:

In this work, PG-NLNN is specifically designed for soil moisture forecasting at multiple depths in the soil profile, as depicted in Figure 2. We have decoupled the non-local operations rooted in the characteristics of soil water transport mechanisms. Taking into account gravity, capillary action, and soil water retention, this physics-guided non-local operation comprises four factors that influence soil moisture at a fixed depth: upper boundary conditions, upper soil moisture, soil moisture within the same depth at the previous time step, and soil moisture at lower depths. These components collectively coordinate and contribute to the soil moisture prediction at the specific depth.

When analyzing the soil moisture at $i_{th}$ depth, denoted as $\boldsymbol{y}_i$, its dynamics are influenced by several factors: upper boundary conditions represented by $\boldsymbol{x}_0$, upper soil moisture state at the previous time step, $\boldsymbol{x}_u$, (where $u < i$, primarily donated by gravity), lower soil moisture $\boldsymbol{x}_l$, (where $l < i$, mainly affected by capillary), and the soil moisture at the same depth from the previous time step, $\boldsymbol{x}_i$. Since these four components are motivated by diverse physical mechanisms, they are defined in distinct forms within the non-local operation.

Before proceeding to the subsections, we provide a brief introduction to fully-connected neural networks (FNNs) that are utilized in the following sections. A two-layer fully-connected neural network can be defined as follows:

$$FNN(\boldsymbol{x}_{input}) = a(W_2(a(W_1\boldsymbol{x}_{input} + b_1) + b_2)$$

(6)

where $a$ denotes the activation function, and $W_L$ and $b_L$ represent the weight matrices and bias parameters to be learned in the $L_{th}$ layer, respectively, where $L = 1,2$. $\boldsymbol{x}_{input}$ denotes the input vector of an FNN. This two-layer FNN is commonly used in our research. In this architecture, we adopt the hyperbolic tangent function as the activation function $a$.

The effect of upper boundary conditions on soil moisture at depth $\boldsymbol{z}_i$ is described by the function, $f_0(\boldsymbol{x}_i, \boldsymbol{x}_j)$, which corresponds to three factors: $\boldsymbol{x}_0$, the meteorological factor; $\boldsymbol{x}_i$, the soil moisture at depth $\boldsymbol{z}_i$ from the previous time step; and $\boldsymbol{z}_i$, the depth of the concerned soil moisture. $\boldsymbol{z}_i$ denotes the $i_{th}$ depth in the depth vector $\boldsymbol{z} = [z_0, z_1, \ldots, z_n]^{\mathrm{T}}$, which corresponds to the input soil moisture data $\boldsymbol{sm}^t$. We utilize a two-layer FNN to describe this relationship:



$$f_0(\boldsymbol{x}_i, \boldsymbol{x}_j) = FNN_0(\boldsymbol{x}_0, \boldsymbol{x}_i, \boldsymbol{z}_i), j = 0 \tag{7}$$

In considering the impacts of soil moisture in the upper layers and lower layers on soil moisture at depth $\boldsymbol{z}_i$, we propose $f_u(\boldsymbol{x}_i, \boldsymbol{x}_j)$ and $f_l(\boldsymbol{x}_i, \boldsymbol{x}_j)$ to calculate the effects. Both functions are determined by the disparity in soil moisture content $(\boldsymbol{x}_i - \boldsymbol{x}_j)$, the intrinsic soil moisture $\boldsymbol{x}_i$, and the distance between two positions $(\boldsymbol{z}_i - \boldsymbol{z}_j)$. As previously stated, two two-layer FNNs are employed in this section:

$$f_u(\boldsymbol{x}_i, \boldsymbol{x}_j) = FNN_u(\boldsymbol{x}_i - \boldsymbol{x}_j, \boldsymbol{x}_i, \boldsymbol{z}_i - \boldsymbol{z}_j), i > j \tag{8}$$

$$f_l(\boldsymbol{x}_i, \boldsymbol{x}_j) = FNN_l(\boldsymbol{x}_i - \boldsymbol{x}_j, \boldsymbol{x}_i, \boldsymbol{z}_i - \boldsymbol{z}_j), i < j \tag{9}$$

Additionally, we utilize relative position encodings to describe the soil water retention effect:

$$f_r(\boldsymbol{x}_i, \boldsymbol{x}_j) = r\_score_{ij}, i = j \tag{10}$$

where the relative position score $r\_score_{ij}$ is utilized for the water retention effect of soil moisture at a specific depth across two adjacent time steps. It can be calculated in Equation (4). Consequently, our position encoding matrix $A_{PG}^K$ is a diagonal matrix comprising $(n + 1)$ distinct elements, which needs to be learned through training:

$$A_{PG} = \begin{pmatrix} \boldsymbol{a}_{0,0} & \cdots & 0 \\ \vdots & \ddots & \vdots \\ 0 & \cdots & \boldsymbol{a}_{n,n} \end{pmatrix} \tag{11}$$

According to the above, the impact on soil moisture at a fixed depth is harmoniously coordinated and integrated through the four components mentioned earlier, as illustrated in Figure 2. Therefore, the physics-guided non-local operation for soil moisture dynamics simulation can be defined as follows:

$$f(\boldsymbol{x}_i, \boldsymbol{x}_j) = e^{f_0(x_i, x_j)/N + f_u(x_i, x_j)/N + f_l(x_i, x_j)/N + f_r(x_i, x_j)/\sqrt{d_k}} \tag{12}$$

$$\mathcal{C}(\boldsymbol{x}) = \sum_{\forall j} f(\boldsymbol{x}_i, \boldsymbol{x}_j) \tag{13}$$

where $N$ is the number of positions in $\boldsymbol{x}$, $\sqrt{d_k}$ denotes the scale factor. Then $\boldsymbol{y}_i$ can be calculated using Equation (1). All operation heads execute similar operations in this model. $W_q$ utilized for $r\_score$ computing and $W_g$ in $g(\boldsymbol{x}_j)$ are still unique in each head. The parameters of the FNNs are shared across non-local operation heads.

**2.3 Boundary processing**



In our soil moisture prediction task, the impact of the upper boundary conditions on soil moisture is partially simulated by an LSTM module (Hochreiter & Schmidhuber, 1997), as illustrated in Figure 2. We have selected six meteorological variables to characterize the influence of these upper boundary conditions: precipitation (P), air temperature (AT), long-wave radiation (LR), short-wave radiation (SR), relative humidity (RH), and wind speed (WS). These variables, denoted as $\boldsymbol{ub}^t =$

$[P^t, AT^t, LR^t, SR^t, RH^t, WS^t]^{\mathrm{T}}$, are closely associated with the infiltration and evapotranspiration processes. Accounting for the delayed effect of meteorology on soil moisture, the input of our LSTM comprises data from two time steps. Following LSTM processing, the impact of the upper boundary conditions takes the form of $sm_0^t$, which is subsequently utilized in non-local operations in conjunction with the input soil moisture data $[sm_1^t, sm_2^t, \dots, sm_{n-1}^t, sm_n^t]^{\mathrm{T}}$ within the soil profile. The operation of

an LSTM can be summarized as follows:

$$\boldsymbol{i}^t = a(W_i \cdot [\boldsymbol{h}^{t-1}, \boldsymbol{ub}^t] + b_i) \tag{14}$$

$$\boldsymbol{f}^t = a(W_f \cdot [\boldsymbol{h}^{t-1}, \boldsymbol{ub}^t] + b_f) \tag{15}$$

$$\boldsymbol{o}^t = a(W_o \cdot [\boldsymbol{h}^{t-1}, \boldsymbol{ub}^t] + b_o) \tag{16}$$

$$\widetilde{\boldsymbol{C}}^t = tanh(W_c \cdot [\boldsymbol{h}^{t-1}, \boldsymbol{ub}^t] + b_c) \tag{17}$$

$$\boldsymbol{c}^t = \boldsymbol{f}^t \cdot \boldsymbol{c}^{t-1} + \boldsymbol{i}^t \cdot \widetilde{\boldsymbol{C}}^t \tag{18}$$

$$\boldsymbol{h}^t = \boldsymbol{o}^t \cdot tanh(\boldsymbol{c}^t) \tag{19}$$

where $W_i$ and $b_i$, $W_f$ and $b_f$, $W_o$ and $b_o$ denote the deep learning parameters for the input gate, forget gate, and the output gate, respectively; $W_c$ and $b_c$ are the parameters for cell state updating; in addition, $\boldsymbol{i}^t$, $\boldsymbol{f}^t$ and $\boldsymbol{o}^t$ are the input gate, forget gate, and output gate at time $t$, respectively, and $\boldsymbol{c}^t$ is the memory cell state; $\boldsymbol{h}^t$ represents the hidden state; $a$ is the sigmoid activation function.

Through sequential processing, the last hidden state $\boldsymbol{h}^t$ in the output $[\boldsymbol{h}^{t-1}, \boldsymbol{h}^t]$ derived from input $[\boldsymbol{ub}^{t-1}, \boldsymbol{ub}^t]$, which encodes the upper boundary effect over two time steps, is adopted as the $sm_0^t$. In this study, the lower boundary conditions are disregarded due to the obstacles in observation.

**2.4 Training Strategies**



The objective of our model is to simultaneously predict soil moisture at multiple depths for the next

time step. To achieve this, we define the loss function as the sum of squared errors between the model

predictions and the corresponding ground truth of soil moisture content at different depths. The model is

trained by minimizing this loss function:

$$\mathcal{L} = \sum_{t=0}^{B} \sum_{i=1}^{n} \left( sm_i^{t+1\prime} - sm_i^{t+1} \right)^2 \tag{20}$$

where $n$ denotes the number of concerned soil moisture depths, and $B$ is the training batch size, which

is set to 100 in this study.

In this work, the collected data is divided into training, validation, and test sets in a time-ordered ratio

of 6:2:2. For training, we employ the Adam optimizer (Kingma & Ba, 2015) with a learning rate of 0.001.

The models are trained for a minimum of 2500 epochs, with 20 batches in each epoch. The validation set

is utilized to select the best model and mitigate overfitting. Subsequently, the test set is then employed

to evaluate the performance of the models. Each result is computed based on 10 replicates with different

initializations. Regarding the model hyperparameter settings, in the non-local neural network, we set

$d_k = 10, d_v = 16,$ and $n_{head} = 10$, where $d_k$ and $d_v$ represents the dimensions of the key and query

embeddings, respectively. $n_{head}$ denotes the number of non-local heads. The LSTM consists of two

stacked blocks, each configured with a hidden layer of 20 neurons. In the FNN adopted for PG-NLNN,

we utilize 10 neurons in each hidden layer.

**3. Data Descriptions**

In our study, synthetic soil moisture data is generated to investigate the interpretability of these NLNN

models. Additionally, we utilize the selected in-situ soil moisture data to assess the accuracy and

practicability of our models.

**3.1 Synthetic Data Description**

The synthetic data are generated using the ROSS method (P J Ross, 2003; Peter J Ross, 2006). The

Ross method is a rapid, non-iterative numerical scheme for soil moisture forward modeling. In our

simulation, we create soil moisture content data for a 100 cm soil column with 1 cm intervals. For

boundary conditions, the daily reference evapotranspiration (ET0) is calculated with the FAO Penman-

Monteith method (Allen et al., 1998) in Wuhan coordinates to generate the synthetic data. The daily time



series data of precipitation and calculated evapotranspiration are shown in Figure 3. The lower boundary

condition is set as free drainage, and the initial moisture content of the soil column is set as 0.10. We

generate three years of time series soil moisture data for this research.

In this section, we design four virtual cases of different configurations to investigate model

interpretability, including homogeneous soil, heterogeneous soil, two-layered soil, and soil with root

water uptake scenarios, as represented in Figure 4. When generating synthetic data in the case with root

water uptake, the root depth is set to 50cm, and root density is vertically distributed evenly. Detailed soil

property settings are given in Appendix A. Besides, we assess the adaptability across different time scales

and observation locations using the available data.

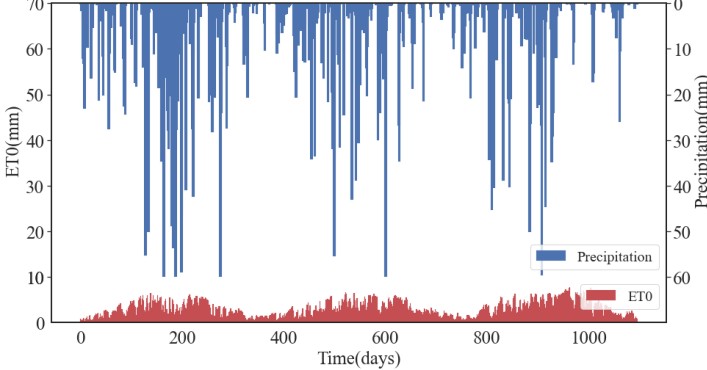

**Figure 3.** Daily time series precipitation and reference evapotranspiration data calculated at Wuhan coordinate for

generating synthetic data.

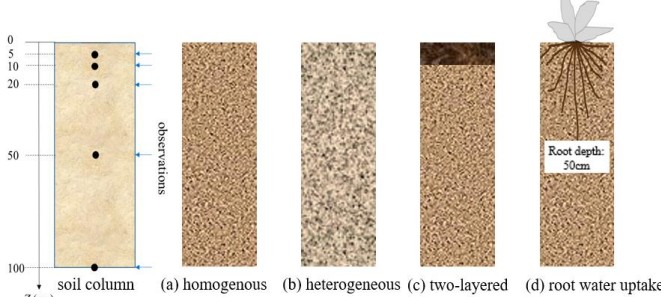

**Figure 4.** The virtual cases design, with homogeneous soil (a), heterogeneous soil (b), two-layered soil (c), and

homogeneous soil with root water uptake (d).





### 3.2 In-situ Data Description

To comprehensively evaluate the proposed NLNN models, we carefully select soil moisture content observations from twenty sites within the International Soil Moisture Network (ISMN) (https://ismn.geo.tuwien.ac.at/en/). These sites are chosen based on geographical locations, soil textures,

and land cover types. Detailed information for the selected sites is presented in Table 1, and their spatial locations are illustrated in Figure 5. These carefully selected sites encompass 16 soil types and 6 land cover species, providing a diverse range to assess the model's performance and its ability to adapt to complex soil situations. At each site, in-situ observations are required to include soil moisture observations at 5 standard depths (0.05m, 0.10m, 0.20m, 0.50m, 1.00m).


**Table 1**. Summary of main characteristics of twenty selected sites.

| Number | Site | Sand | Silt | Clay | Land cover | Period | Lat. | Lon. |
|---|---|---|---|---|---|---|---|---|
| 1 | Kingston-1-W | 85 | 10 | 5 | Grassland | 2012-2023 | 41.48 | -71.54 |
| 2 | Monahans-6-ENE | 83 | 6 | 11 | Shrub cover | 2010-2022 | 31.62 | 102.81 |
| 3 | Necedah-5-WNW | 83 | 11 | 6 | Grassland | 2009-2022 | 44.06 | -90.17 |
| 4 | Shadow Mtns | 79 | 10 | 11 | Shrub cover | 2013-2017 | 35.47 | -115.72 |
| 5 | Falkenberg | 73 | 21 | 6 | Cropland, rained | 2003-2020 | 52.17 | 14.12 |
| 6 | Kenai-29-ENE | 54 | 38 | 8 | Shrub cover | 2012-2023 | 60.72 | -150.45 |
| 7 | AAMU-jtg | 53 | 22 | 25 | Grassland | 2010-2022 | 34.78 | -86.55 |
| 8 | Darrington-21-NNE | 53 | 22 | 25 | Tree cover | 2013-2019 | 48.54 | -121.45 |
| 9 | Palestine-6-WNW | 49 | 27 | 24 | Grassland | 2009-2013 | 31.78 | -95.72 |
| 10 | Cullman | 49 | 27 | 24 | Mosaic Cropland | 2006-2022 | 34.20 | -86.80 |
| 11 | Cape-Charles | 49 | 27 | 24 | Herbaceous cover | 2011-2022 | 37.29 | -75.93 |
| 12 | LittleRiver | 47 | 30 | 23 | Grassland | 2005-2020 | 31.50 | -83.55 |
| 13 | Montrose-11-ENE | 43 | 35 | 22 | Tree cover | 2010-2023 | 38.54 | -107.69 |
| 14 | Coshocton-8-NNE | 41 | 39 | 20 | Grassland | 2009-2016 | 40.37 | -81.78 |
| 15 | Bodega-6-WSW | 39 | 38 | 23 | Grassland | 2011-2023 | 38.32 | -123.08 |
| 16 | Goodwell-2-SE | 36 | 41 | 23 | Grassland | 2010-2022 | 36.57 | -101.61 |
| 17 | Riley-10-WSW | 36 | 41 | 23 | Shrub cover | 2011-2021 | 43.47 | -119.69 |
| 18 | Joplin-24-N | 35 | 41 | 24 | Grassland | 2010-2020 | 37.43 | -94.58 |
| 19 | Weslaco | 34 | 45 | 21 | Cropland, rained | 2017-2021 | 26.16 | -97.96 |
| 20 | UpperBethlehem | 32 | 38 | 30 | Herbaceous cover | 2008-2010 | 17.72 | -64.80 |



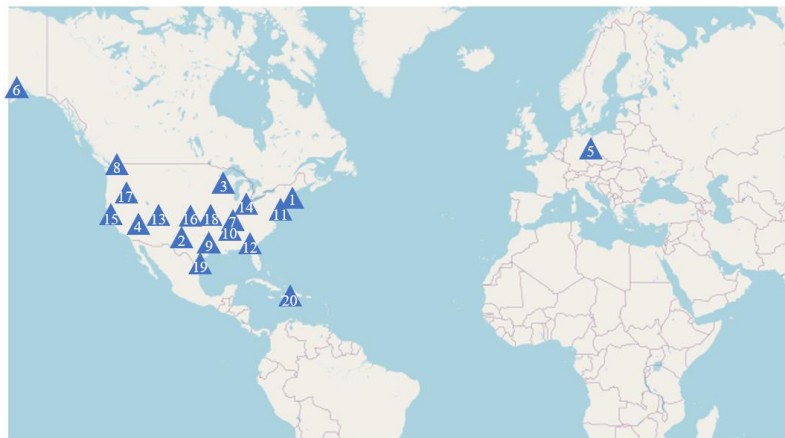

**Figure 5**. The spatial locations of twenty selected sites. The numbers on the sites correspond to the serial numbers in Table 1.


The meteorological inputs for our models include precipitation, atmospheric temperature, long-wave radiation, short-wave radiation, wind speed, and relative humidity, as mentioned above. These meteorological data are sourced from the NASA Prediction of Worldwide Energy Resources project (https://power.larc.nasa.gov/). Detailed information about this can be found at
(https://power.larc.nasa.gov/docs/methodology/data/sources/). Unfortunately, due to challenges in obtaining groundwater level observations, changes in the lower boundary conditions are not considered in this study.

**4. Results and discussions**

In this study, we systematically examine and analyze our models from three perspectives. Initially, we
assess the essential capabilities of models, including accuracy and uncertainty, using both synthetic data and in-situ observations. Subsequently, we apply simulated soil moisture data under diverse virtual scenarios to evaluate our model's interpretability and its ability to provide qualitative interpretations depicting soil moisture interaction mechanisms across diverse depths within the profile. Finally, we investigate the impacts of varying temporal scales, noise levels, and observation locations on our non-
local neural networks.





To explore the forecasting ability of our models over time series, we examine predictions for 1, 3, and 7 days ahead at selected sites, as well as 1, 3, 7, and 15 days ahead for simulated data. We generate predictions iteratively. The evaluation standards in this work comprise the mean absolute error (MAE) and the root mean square error (RMSE). Both MAE and RMSE quantify the deviation between the

predictions and the ground truth. However, RMSE exhibits greater sensitivity to outliers due to its squaring of deviations, which amplifies the impact of extreme values, while MAE offers a smoother average error value. These metrics are calculated as follows:

$$\text{MAE} = \frac{\sum_{i=1}^{N_s} |T_i - \hat{T}_i|}{N_s} \tag{21}$$

$$\text{RMSE} = \sqrt{\frac{\sum_{i=1}^{N_s} (T_i - \hat{T}_i)^2}{N_s}} \tag{22}$$

where $\hat{T}_i$ and $T_i$ represent the predictions and the ground truth, respectively; $\bar{T}_i$ is the average of the ground truth; $N_s$ is the test sample size. Here, $T$ denotes the soil moisture content [%] which needs to

be calculated.

When conducting uncertainty analysis, evaluating confidence bounds becomes challenging because most deep learning neural networks are essentially deterministic models. To address this, many researchers utilize the bootstrap aggregating (bagging) method (Breiman, 1996) to analyze model predictive uncertainty (Kornelsen & Coulibaly, 2014). The bagging method involves training multiple

neural network models using subsets of the training set, all with identical architecture. To create the training subset for each model, a statistical bootstrap approach is employed. For each subset, we randomly select individual input vectors from the entire training set with replacement, ensuring that each subset contains the same number of elements as the entire training set. After training, we obtain an ensemble of trained models, each trained with a unique training subset. The final output and uncertainty

estimates are then derived from the mean and standard deviation of this ensemble.

To explore the impact of noise on our models using the synthetic data, we apply the zero-mean Gaussian noise with a variance of 1:

$$\dot{\theta} = \theta + \eta * \mathcal{N}(0,1), \tag{23}$$





where $\dot{\theta}$ is the volumetric soil moisture content with noise [%], and $\theta$ is the synthetic volumetric soil moisture content. Three noise levels are tested ($\eta$ = 0.5, 1.0, 2.0) in this work.

In our investigation of model interpretability, the visualized non-local weight maps generated from the output play a crucial role as evaluation standards. These weight maps may provide qualitative interpretations depicting intricate mechanisms of soil water dynamics. The color brightness on the weight distribution map signifies the level of interaction strength among upper boundary conditions and soil moisture across different depths. Therefore, analyzing the weight matrix map is essential for gaining

insights into the learning mechanisms of our NLNN models.

### 4.1 Interpretability analysis

Before the models can be applied to real-world scenarios, their stability and interpretability must first be analyzed. In this section, we explore the interpretability of the NLNN models by designing several scenarios that generate synthetic data. These simulated cases primarily involve variations in soil

properties, including homogeneous soil, heterogeneous soil, two-layered soil, and soil with root water uptake scenarios. With the synthetic data, we investigate the model interpretability through the weight matrix maps and delve into their learning mechanisms across diverse scenarios.

**Table 2.** The MAE [%] values for 1, 3, 7, and 15-day forecasts of the proposed PG-NLNN model and SA-NLNN model at 5 depths under four designed scenarios.

| Depth/m | PG-NLNN | | | | | | | | | | | | | | | |
|---|---|---|---|---|---|---|---|---|---|---|---|---|---|---|---|---|
| | homogeneous | | | | heterogeneous | | | | two-layered | | | | root water uptake | | | |
| | 1d | 3d | 7d | 15d | 1d | 3d | 7d | 15d | 1d | 3d | 7d | 15d | 1d | 3d | 7d | 15d |
| 0.05 | **0.235** | **0.327** | **0.433** | **0.539** | **0.259** | **0.372** | **0.510** | **0.652** | 0.449 | 0.680 | 0.945 | **1.170** | 0.494 | **0.698** | **0.938** | **1.212** |
| 0.10 | 0.313 | 0.451 | 0.627 | **0.788** | 0.306 | 0.431 | 0.593 | **0.749** | 0.521 | 0.745 | 0.995 | **1.191** | 0.382 | 0.518 | 0.677 | **0.925** |
| 0.20 | 0.342 | 0.533 | 0.776 | 1.016 | 0.305 | 0.488 | 0.736 | 0.971 | **0.433** | **0.649** | **0.901** | **1.179** | 0.565 | 0.795 | 1.031 | **1.208** |
| 0.50 | 0.235 | 0.357 | 0.545 | 0.782 | **0.253** | 0.399 | 0.630 | 0.952 | 0.334 | 0.518 | 0.774 | 1.098 | 0.379 | 0.641 | 1.127 | 1.752 |
| 1.00 | **0.203** | **0.312** | **0.445** | **0.647** | **0.244** | 0.397 | 0.618 | 0.934 | 0.368 | 0.625 | 0.969 | 1.329 | **0.278** | **0.470** | **0.788** | **1.288** |
| Depth/m | SA-NLNN | | | | | | | | | | | | | | | |
| | homogeneous | | | | heterogeneous | | | | two-layered | | | | root water uptake | | | |
| | 1d | 3d | 7d | 15d | 1d | 3d | 7d | 15d | 1d | 3d | 7d | 15d | 1d | 3d | 7d | 15d |
| 0.05 | 0.328 | 0.470 | 0.686 | 1.039 | 0.363 | 0.524 | 0.750 | 1.840 | **0.327** | **0.505** | **0.836** | 2.210 | 0.536 | 0.918 | 2.150 | 6.702 |
| 0.10 | **0.249** | **0.375** | **0.580** | 0.957 | **0.220** | **0.314** | **0.477** | 0.851 | **0.390** | **0.569** | **0.808** | 1.465 | **0.322** | **0.447** | **0.675** | 1.480 |
| 0.20 | **0.262** | **0.366** | **0.519** | **0.820** | 0.292 | 0.389 | 0.482 | 0.648 | 0.487 | 0.696 | 0.945 | 1.350 | **0.379** | **0.546** | **0.775** | 1.861 |
| 0.50 | **0.209** | **0.291** | **0.414** | **0.566** | 0.265 | **0.337** | **0.431** | **0.623** | **0.327** | **0.483** | **0.708** | **1.018** | **0.344** | **0.485** | **0.687** | 1.502 |
| 1.00 | 0.245 | 0.376 | 0.575 | 0.807 | 0.282 | 0.430 | 0.640 | 0.941 | **0.336** | **0.530** | **0.810** | **1.250** | 0.297 | 0.482 | 0.820 | 1.748 |





Table 2 summarizes the MAE values for 1, 3, 7, and 15-day forecasts across four simulated cases, while Figure 6 displays the RMSE results and their variance over ten trainings. According to the MAE results, the performance of both models is comparable. However, the physics-guided model PG-NLNN exhibits lower variance and maintains greater stability in RMSE, especially in the 15-day prediction task

of soil moisture dynamics. This indicates that the PG-NLNN may be more effective in handling specific extreme situations. The integration of physics guidance proves crucial in ensuring model stability. To illustrate how forecasts diverge over time, we have plotted Figure 7, which shows the 30-day autoregressive soil moisture prediction. The results show that both our NLNN models closely match the observations from the synthetic data and remain stable over time, without incurring significant errors as

time progresses.

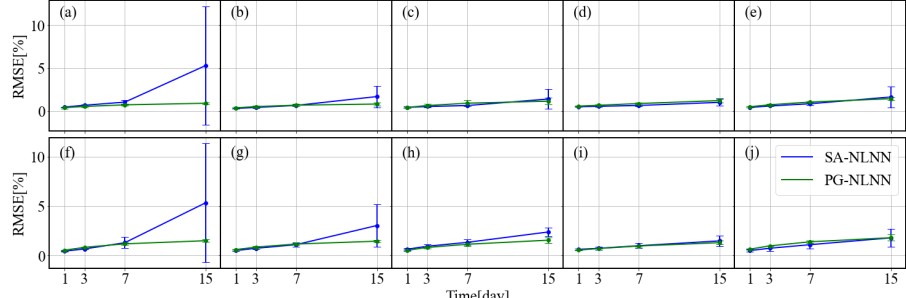

**Figure 6.** The RMSE results for 1, 3, 7, and 15-day for heterogeneous soil(a-e), and two-layered soil (f-j). The error bar indicates the standard deviations of the RMSE, which are computed via ten training replicates.


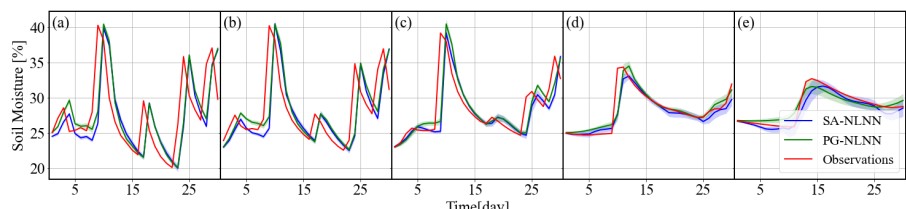

**Figure 7**. The autoregressive 30-day predicted time series of PG-NLNN, SA-NLNN of soil moisture for heterogeneous soil at 5 depths: 0.05m(a), 0.10m(b), 0.20m(c), 0.50m(d), 1.00m(e). The shaded region represents the confidence interval of the models, spanning 1 standard deviation.




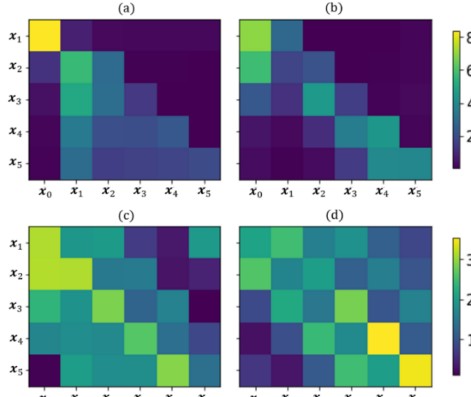

**Figure 8.** The non-local weight maps in two-layered simulated stratified soil scenarios through PG-NLNN (a) loam above sand (b) sand above loam, and SA-NLNN (c) loam above sand (d) sand above loam.

Figure 8 depicts the weight matrix maps generated by PG-NLNN and SA-NLNN models for two-layered soil scenarios. The elements at position $(i,j)$ represent the impact of soil moisture at depth $z_j$ at the previous time on soil moisture content at depth $z_i$. Notably, when $j = 0$, it signifies the influence of upper boundary conditions on soil moisture across various depths. The brightness level corresponds to the strength of this influence, with higher brightness indicating a stronger impact. Specifically, we use

the two-layered soil case and switch the soil properties of the upper and lower layers to observe the changes in the non-local weight matrices. The saturated hydraulic conductivity of the two soil types varies significantly, with distinct characteristics influencing water transport and drainage, as recorded in Appendix A. Figure 8 presents the weight matrix maps generated through PG-NLNN and SA-NLNN, which is the $\frac{f(x_i, x_j)}{\mathcal{C}(x)}$ calculated through non-local operations. The PG-NLNN weight map reveals the

interaction strength of soil moisture at different depths within the profile.

Some soil structural information, such as stratification, can be reflected from the soil moisture interactions. In the scenario where sand is beneath loam, the upper loam layer gradually releases water, while the lower sand layer quickly absorbs it. The water released from the loam layer can quickly reach various depths of the sand layer below. Consequently, the soil moisture in the lower layers is primarily

influenced by the upper loam layer. As shown in Figure 8(a), the moisture in the lower layer (0.10m,





0.20m, 0.5m, 1.0m) is notably influenced by the moisture at 0.05m. Conversely, when sand is above the loam, the upper sand layer rapidly drains water, and the lower loam layer more effectively absorbs and retains it. As a result, the water from the upper sand is absorbed and held by the lower loam. Therefore, soil moisture in the lower layer is mainly affected by the adjacent upper layer, as shown in Figure 8(b).

However, the weight map of the SA-NLNN model appears slightly chaotic, as depicted in Figure 8(c) and (d). It indicates that the SA-NLNN model, lacking physical guidance, tends to learn the incorrect relationships. This highlights that incorporating a suitable structural design guided by physics laws can be a valuable addition. Enhancements in PG-NLNN not only improve adaptability and interpretability concerning soil properties but also contribute to overall stability.

As a result, both NLNN models achieve satisfactory soil moisture forecasts in the simulated scenarios. Furthermore, the models have advanced the interpretability of machine learning through non-local weight matrix maps. Notably, PG-NLNN offers more reliable descriptions of soil properties via these visualizations, highlighting the importance of physics guidance.

**4.2 Performance evaluation**

In this section, we evaluate the performance of the SA-NLNN and PG-NLNN models using in-situ observations from twenty ISMN sites. We benchmark our soil moisture prediction tasks against the LSTM model, widely used in time series forecasting (Datta & Faroughi, 2023; Ding et al., 2019; Siami-Namini et al., 2019). Specifically, the LSTM model takes two forms tailored for different data processing approaches: LSTM_4, which utilizes input data from the previous four time steps to predict soil moisture content at the next time step. It follows a configuration similar to that in previous work (Y. Wang, Shi, Hu, Hu, et al., 2023). These predictions rely on modeling temporal dependencies. In contrast, LSTM_1, aligned with our NLNN model structure depicted in Figure 2, replaces non-local operations with LSTM modules. It represents the predictive capabilities achievable by a single-time-step LSTM. The performance of all models is evaluated at five different depths (0.05m, 0.1m, 0.2m, 0.5m, 1.0m). Notably,

our NLNN models predict soil moisture for all five depths simultaneously, whereas LSTM models each depth separately. When comparing our models with physical models, the inherent methodological differences between machine learning and physical models make fair and direct comparisons with



standard physics-based modeling particularly challenging. We therefore limit our comparison to a preliminary assessment in Appendix B.

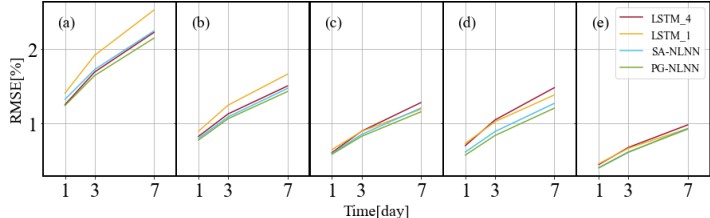

**Figure 9.** The average RMSE comparisons between LSTM_4, LSTM_1, SA-NLNN, and PG-NLNN across twenty research sites at 5 depths: 0.05m(a), 0.10m(b), 0.20m(c), 0.50m(d), 1.00m(e).

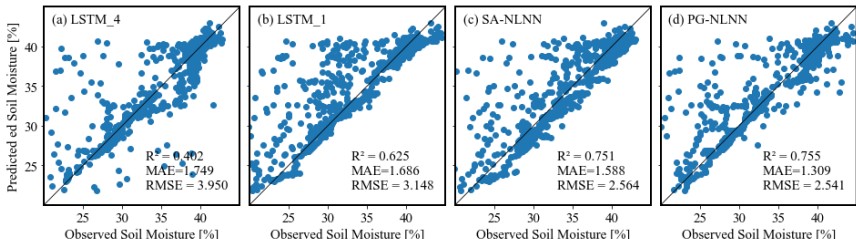

**Figure 10.** Scatter plots of the soil moisture observations and 7-day predictions generated from (a) LSTM_4, (b) LSTM_1, (c) SA-NLNN, and (d) PG-NLNN at UpperBethlehem.

**Table 3.** The MAE [%] values for 1, 3, and 7-day forecasts across the four models across twenty research sites at 5 distinct depths, based on ten repeated trainings.

| depth/m | MAE | | | | | | | | | | | |
|---|---|---|---|---|---|---|---|---|---|---|---|---|
| | PG-NLNN | | | SA-NLNN | | | LSTM_4 | | | LSTM_1 | | |
| | 1d | 3d | 7d | 1d | 3d | 7d | 1d | 3d | 7d | 1d | 3d | 7d |
| 0.05 | **0.391** | **0.600** | **0.893** | 0.440 | 0.666 | 0.979 | 0.737 | 1.074 | 1.515 | 0.808 | 1.203 | 1.713 |
| 0.10 | **0.392** | **0.603** | **0.900** | 0.431 | 0.659 | 0.972 | 0.498 | 0.726 | 1.027 | 0.506 | 0.771 | 1.113 |
| 0.20 | 0.397 | 0.607 | 0.900 | 0.431 | 0.648 | 0.947 | **0.356** | 0.558 | 0.844 | 0.357 | **0.547** | **0.787** |
| 0.50 | **0.392** | **0.601** | **0.896** | 0.432 | 0.648 | 0.962 | 0.405 | 0.632 | 0.955 | 0.403 | 0.620 | 0.909 |
| 1.00 | 0.394 | 0.602 | 0.885 | 0.422 | 0.641 | 0.943 | 0.245 | 0.386 | 0.597 | **0.243** | **0.385** | **0.592** |

Table 3 displays the MAE values across twenty selected sites, considering forecasts for 1, 3, and 7 days from the four models at five distinct depths. These results are derived from ten repeated trainings, and the corresponding RMSE results are presented in Figure 9. From MAE results, we observe that both

LSTM_1 and LSTM_4 perform well in deep soil moisture predictions. Meanwhile, our proposed NLNN

models consistently demonstrate superior accuracy at depths from 0.05m to 0.5m. Regarding RMSE, the

PG-NLNN model stands out as the best model in most situations. This indicates that the PG-NLNN

model is more adept at managing specific extreme conditions compared to LSTM-based models. Figure

10 depicts the correlation between the 7-day soil moisture predictions and observations of the test set for

LSTM-4, LSTM-1, SA-NLNN, and PG-NLNN. The density of scatter plots serves as an indicator of

model reliability (Datta & Faroughi, 2023). The PG-NLNN model exhibits superior performance in soil

moisture prediction compared to the other models, suggesting the stability of our model over longer

prediction periods. Through the comparison of PG-NLNN and SA-NLNN, the importance of using soil

water transport mechanisms to guide the design of decoupled non-local operations is evident.

Nevertheless, a limitation of the proposed NLNN models lies in their forecasts for moisture content at

1.0m. This limitation could be attributed to the absence of consideration for lower boundary conditions

in our study.

Regarding how NLNN model predictions change over time, Figure 11 displays the autoregressive 24-

day predicted time series soil moisture data for the NLNN models across three sites: Falkenberg, Cape-

Charles, and Goodwell. The shaded region represents the confidence interval of the NLNN models,

spanning 1 standard deviation. The LSTM-based models exhibit relatively greater uncertainty in

predictions. However, it is evident that both models perform satisfactorily and stably, with the proposed

PG-NLNN model being closer to the observations. Considering the temporal accumulation of

autoregressive errors in extended soil moisture forecasting, we provide additional long-term prediction

results in Appendix B for comprehensive evaluation.






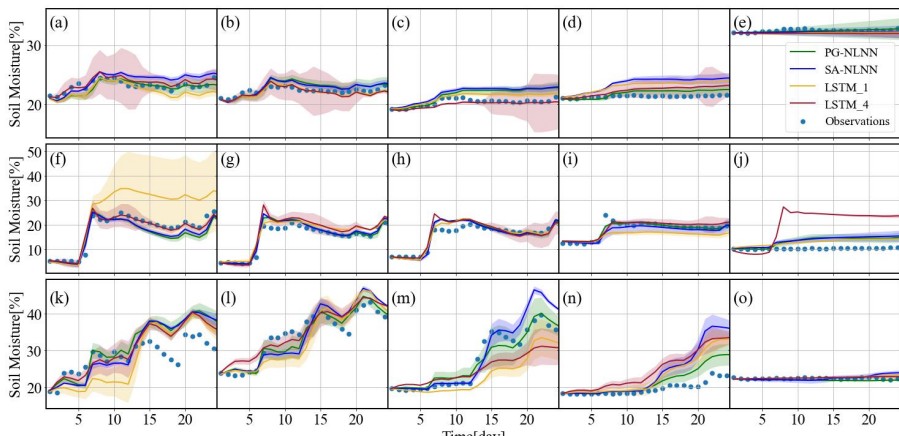

**Figure 11**. The autoregressive 24-day predicted soil moisture time series of 5 depths with LSTM_1, LSTM_4, PG-NLNN and SA-NLNN at Falkenberg (a-e), Cape-Charles (f-j), and Goodwell (k-o). The shaded region represents the confidence interval of the models, spanning 1 standard deviation.


According to section 4.1, the non-local weight maps can be related to the soil properties, demonstrating the interpretability of the model. In real-world cases, even with limited soil information from the site in Table 1, we can combine the weight maps with the measured soil texture data for our analysis. Figure 12 illustrates the non-local weight matrix maps for the Falkenberg, Cape-Charles, and UpperBethlehem sites, generated by the PG-NLNN model. These maps remain stable during repeated training, with discernible variations among the three sites. They offer qualitative interpretations related to soil properties. In Figure 12(a), it is seen that at Falkenberg site, soil moisture at different depths is primarily influenced by upper boundary conditions and upper layer soil moisture. Figure 12(b) shows that at Cape-Charles site, soil moisture is mainly affected by upper boundary conditions and soil moisture at the same depth from the previous time step. Figure 12(c) depicts the strong soil water retention effect at UpperBethlehem site, soil moisture is mainly related to its own state at the previous time step. By combining Table 1, we can see that the non-local weight maps are consistent with the soil texture information. From Falkenberg to UpperBethlehem site, as the soil texture changes from sandy to clay, the learnt water retention capacity in Figure 12 increases from low to high. Consequently, the non-local weight maps are able to capture different physical mechanisms of different sites from the measurement data.





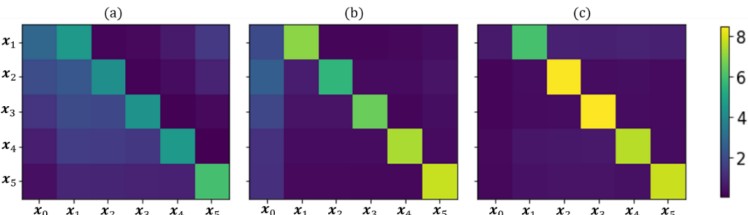

**Figure 12.** The non-local weight maps through the PG-NLNN at three typical sites, (a) Falkenberg, (b) Cape-Charles, and (c) UpperBethlehem.


In summary, based on comparisons with LSTMs using in-situ observations, our models achieve precise and efficient soil moisture predictions across diverse scenarios. Simultaneously modeling soil moisture at different depths in our NLNN models allows for complementary interactions, thereby improving overall accuracy. The proposed PG-NLNN model excels with satisfactory predictions and limited

uncertainty, while also providing qualitative descriptions of the intricate soil properties. The advantages of incorporating physics guidance in non-local operation design are obvious.

**4.3 Effects of the time scales, observation positions, and noise levels**

In addition to model accuracy and interpretability, our non-local neural network exhibits adaptability in prediction tasks across different time scales. In this section, we have conducted tests involving various

time intervals, observation positions, and noise levels. When investigating the PG-NLNN model's performance at the 0.2-day, 0.5-day, and 1-day time intervals within homogenous soil, a subtle difference emerges in the weight map generated by the PG-NLNN model, as illustrated in Figure 13. Despite a decrease in accuracy with longer time intervals, the model consistently achieves satisfactory results. The results reflect the adaptability of the model to diverse time scales.

When the number of observation locations increases to 10 (at depths of 0.05m, 0.1m, 0.2m, 0.3m, 0.4m, 0.5m, 0.6m, 0.7m, 0.8m, 0.9m), the MAE values for soil moisture 1, 3, 7, and 15-day forecasts of the NLNN models across five depths are summarized in Table 4. The uniform augmentation of measurements significantly enhances the prediction accuracy of SA-NLNN, while having minimal impact on the performance of PG-NLNN. This suggests that the physics guidance allows for lower



requirements on soil moisture measurements. In scenarios with uniformly augmented observations, SA-
NLNN may prove more efficient.

In Figure 14, we present the RMSE results for soil moisture predictions at depths of 0.05m, 0.2m, and
1.0m under increasing noise levels. It's worth noting that noise has a more pronounced impact on the soil
moisture predictions in the surface layer. Despite the higher predicted RMSE with increasing noise levels,

both NLNN models display robustness. The PG-NLNN model, in particular, shows greater resistance to
noise levels, consistent with its performance on in-situ soil moisture data.

In conclusion, both the NLNN models achieve accurate and reliable soil moisture predictions under
diverse scenarios. They can adapt to tasks across different time scales. The SA-NLNN performs better
under uniformly distributed observations, while the PG-NLNN demonstrates stronger noise resistance.

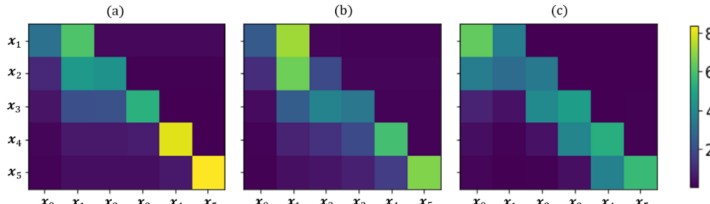


**Figure 13.** The non-local weight maps of the PG-NLNN model at different time scales at 0.2-day (a), 0.5-day (b),
and 1.0-day (c) in the homogenous soil.

**Table 4.** The MAE [%] values for 1, 3, 7, and 15-day forecasts of the proposed PG-NLNN model and SA-NLNN

model at 5 depths with 10 depth measurements under the homogenous soil scenario.

| Depth/m | Homogeneous soil | | | | | | | |
|---|---|---|---|---|---|---|---|---|
| | PG-NLNN | | | | SA-NLNN | | | |
| | 1d | 3d | 7d | 15d | 1d | 3d | 7d | 15d |
| 0.05 | **0.327** | **0.470** | **0.645** | **0.817** | 0.394 | 0.588 | 0.906 | 1.657 |
| 0.10 | 0.280 | 0.407 | 0.602 | **0.825** | **0.250** | **0.350** | **0.535** | 0.892 |
| 0.20 | 0.331 | 0.564 | 0.979 | 1.419 | **0.221** | **0.300** | **0.418** | **0.604** |
| 0.50 | 0.174 | 0.258 | 0.380 | 0.581 | **0.148** | **0.204** | **0.302** | **0.502** |
| 1.00 | **0.108** | 0.180 | 0.300 | 0.493 | 0.118 | **0.174** | **0.259** | **0.460** |



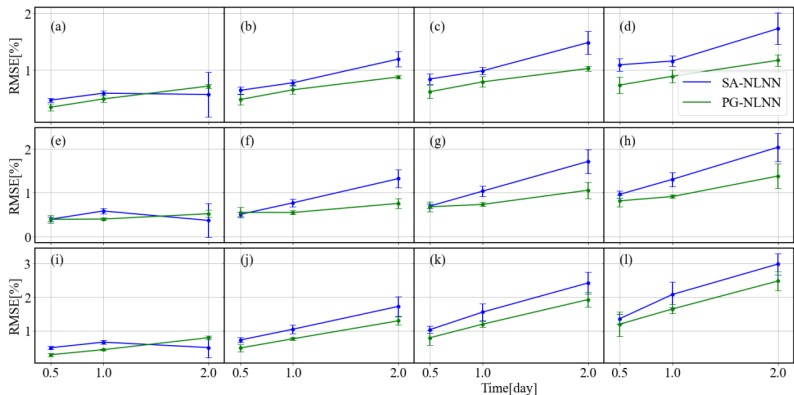

**Figure 14.** The RMSE results for 1, 3, 7, and 15-day at 0.05m(a-d), 0.20m(e-h), and 1.0m(i-l) in the homogenous

soil under increasing noise levels. The error bar indicates the standard deviations of the RMSE, which are computed

via ten training replicates.

## 5. Conclusions

In this study, we employ the deep learning model NLNNs to achieve precise and efficient soil moisture

predictions under diverse scenarios without relying physical assumptions., while providing qualitative

interpretation for complex soil moisture dynamics, such as vertical heterogeneity and inter-layer

connectivity. In light of the accuracy and parameter estimation challenges in physical models, and the

credibility concerns in machine learning models, we have introduced a framework that integrates both

accuracy and mechanistic insight. Our method leverages in-profile soil moisture interactions across

various depths. Consequently, the soil moisture prediction task is reformulated as a single-time-step

prediction task that involves multi-depth soil moisture variables. In this way, we apply the self-attention-

based model SA-NLNN to explore the potential of the NLNN structure. Expanding on this framework,

we disentangle the non-local operation into four components to create the PG-NLNN model, which aligns

with the soil water transport characteristics. By comparing our NLNNs with the LSTM model using

synthetic data and in-situ observations, we demonstrate that both our NLNN models achieve precise and

effective forecasts, providing an alternative possibility for soil moisture simulations. The physics-guided

model PG-NLNN exhibits the best performance and remains stable with low uncertainty. The physics

guidance in non-local operations significantly enhances the model's accuracy and reliability.

Additionally, our proposed models offer qualitative interpretations related to the soil properties. Through the investigation of various virtual scenarios -- including homogeneous soil, heterogeneous soil,

two-layered soil, and soil with root water uptake -- we observe that both the PG-NLNN and SA-NLNN models perform well in different soil conditions. The qualitative interpretations derived from soil moisture data generated by PG-NLNN facilitate descriptions of soil structures. When testing with in-situ data, we find that the PG-NLNN model also provides interpretations consistent with real soil vertical heterogeneity without physical assumptions. This highlights the importance of integrating physics-

guided assistance into model design. Moreover, we have assessed the model's performance under different time scales, observation positions, and noise conditions. The NLNN model demonstrates adaptability to diverse time scales. When measurement positions are evenly distributed, the SA-NLNN model shows significant improvements compared to PG-NLNN, while maintaining high computational efficiency. Besides, both models exhibit robustness to noise, and the physics guidance enhances noise

resistance.

Nevertheless, the model faces challenges that necessitate future improvements. Its training and application are site-specific, limiting its transferability. Further research is required to enhance its applicability across different sites. Specifically, difficulties arise in estimating soil moisture content at deep layers, possibly due to the lack of consideration for the groundwater boundary. Incorporating lower

boundary conditions into the model could address this limitation. Additionally, multi-objective network training may benefit from more effective strategies and more precise loss function designs. Introducing constraints at multiple time steps holds promise for achieving more stable results. Finally, further refinement of the non-local operation may enhance the model's performance. What's more, the proposed network framework is flexible and easily customizable to suit specific requirements, allowing for its

further exploration and extension to address various physical or hydrological problems. We encourage readers to design specialized structures tailored to their respective requirements.

**ACKNOWLEDGEMENTS**



This work was supported by the National Key Research and Development Program of China

(2021YFC3201203) and the National Natural Science Foundation of China (Grant 52179038 and Grant

U2243235).

**CODE/DATA AVAILABILITY**

The    data    and    codes    used    in    this    paper    are    available    on    the    website

(https://doi.org/10.5281/zenodo.10408929).



**Appendix A**

The parameters used to generate the synthetic data are recorded in Table A1 and Table A2:

**Table A1.** The van Genuchten soil hydraulic parameters (van Genuchten, 1980) used for synthetic data generation.

| Case Design | Homogenous soil | Heterogeneous soil | Two-layered soil | Soil with root water uptake |
|---|---|---|---|---|
| $\theta_r$ [−] | 0.078 | 0.078 | 0.078 | 0.078 |
| $\theta_s$ [−] | 0.43 | 0.43 | 0.43 | 0.43 |
| $\alpha$ [$cm^{-1}$] | 3.6 | 3.6 | 3.6 | 3.6 |
| $n$ [−] | 1.56 | 1.56 | 1.56 | 1.56 |
| $K_s$ [$cm\ day^{-1}$] $(0-10cm)$ | 0.250 | Table A2 | 0.250 | 0.250 |
| $K_s$ [$cm\ day^{-1}$] $(10-100cm)$ | 0.250 | Table A2 | 10.49 | 0.250 |
| $l$ [−] | 0.5 | 0.5 | 0.5 | 0.5 |
| Presence of plant | False | False | False | True |


**Table A2.** The soil hydraulic conductivity of the heterogeneous scenario.

| depth[$cm$] | $K_s$ [$cm\ day^{-1}$] | | | | | | | | | |
|---|---|---|---|---|---|---|---|---|---|---|
| $0-10$cm | 0.226 | 0.270 | 0.241 | 0.263 | 0.222 | 0.226 | 0.263 | 0.221 | 0.262 | 0.276 |
| $10-20$cm | 0.230 | 0.226 | 0.217 | 0.226 | 0.249 | 0.203 | 0.229 | 0.196 | 0.207 | 0.202 |
| $20-30$cm | 0.200 | 0.239 | 0.244 | 0.253 | 0.251 | 0.248 | 0.203 | 0.225 | 0.206 | 0.205 |
| $30-40$cm | 0.241 | 0.223 | 0.197 | 0.227 | 0.218 | 0.256 | 0.258 | 0.294 | 0.308 | 0.242 |
| $40-50$cm | 0.242 | 0.155 | 0.177 | 0.184 | 0.218 | 0.230 | 0.225 | 0.211 | 0.207 | 0.252 |
| $50-60$cm | 0.285 | 0.338 | 0.351 | 0.345 | 0.317 | 0.355 | 0.333 | 0.343 | 0.322 | 0.320 |
| $60-70$cm | 0.261 | 0.272 | 0.306 | 0.279 | 0.319 | 0.250 | 0.262 | 0.224 | 0.240 | 0.269 |
| $70-80$cm | 0.269 | 0.300 | 0.276 | 0.250 | 0.267 | 0.233 | 0.240 | 0.249 | 0.207 | 0.233 |
| $80-90$cm | 0.202 | 0.209 | 0.208 | 0.248 | 0.231 | 0.232 | 0.245 | 0.258 | 0.250 | 0.222 |
| $90-100$cm | 0.254 | 0.211 | 0.201 | 0.203 | 0.186 | 0.213 | 0.233 | 0.196 | 0.247 | 0.213 |





**Appendix B**

This section presents a preliminary comparison between the NLNN model and the physics-based soil

moisture model derived from Richards' equation.

The Ross method (P J Ross, 2003; Peter J Ross, 2006) is a rapid, non-iterative numerical scheme for

soil moisture forward modeling based on Richards' Equation. For boundary conditions, the daily

reference evapotranspiration (ET0) is calculated with the FAO Penman-Monteith method (Allen et al.,

1998). We first utilize 10 days of site historical data to invert the site-specific soil hydraulic parameters

($\alpha$, n,$K_s$) through data assimilation with the ensemble Kalman filter (EnKF) method (Evensen, 2003)

within the Ross framework. These parameters are then applied in the Ross method to obtain a fast solution

of one-dimensional Richards' equation, enabling the forecasting of soil moisture dynamics.

In the real-world experiments, we selected three sites: Falkenberg, Cape-Charles, and Goodwell, with

distinctly different soil textures and land covers, as recorded in Table 1 in the manuscript. Figure A1

illustrates the autoregressive 24-day predicted time series soil moisture data for the PG-NLNN model

and Ross-EnKF across these three sites. The MAE results are recorded in Table A3. It is seen that soil

moisture forecasts obtained by PG-NLNN are closer to real observations, compared to the traditional

Ross-EnKF method.

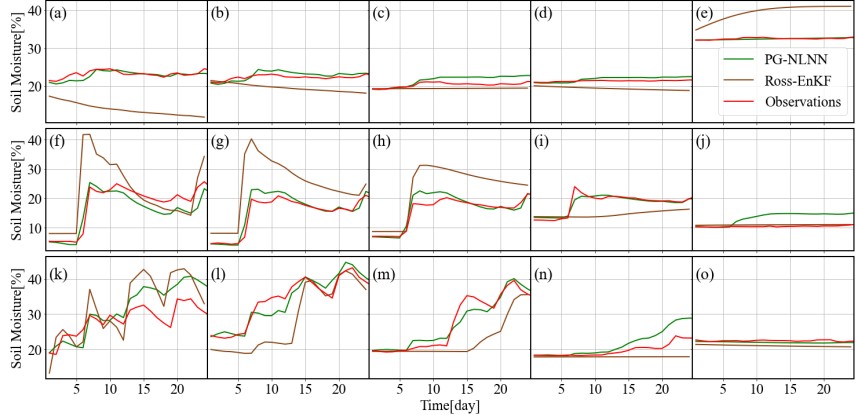


**Figure A1.** The 24-day predicted soil moisture time series of 5 depths with PG-NLNN and Ross-EnKF at Falkenberg

(a-e), Cape-Charles (f-j), and Goodwell (k-o).





However, it should be noted that the data assimilation process in Ross-EnKF did not update soil infiltration parameters, potentially disadvantaging the physical model. What's more, the proposed approaches cannot predict soil moisture at arbitrary depths and times as the physical models. The fundamental differences between machine learning and physical modeling make fair, direct comparisons with standard methods both critical and difficult.

**Table A3.** The MAE [%] values for 24-day forecasts of the proposed PG-NLNN model and Ross-EnKF model

|  | Falkenberg | Cape-Charles | Goodwell |
|---|---|---|---|
| PG-NLNN | 0.681 | 1.766 | 1.998 |
| Ross-EnKF | 4.395 | 5.484 | 3.840 |


Moreover, our machine learning approach exhibits autoregressive error accumulation in long-term soil moisture predictions—a limitation not observed in physics-based modeling. As demonstrated by the 120-day autoregressive forecasts (Figure A2), while model uncertainty gradually accumulates with prediction time, it remains within acceptable bounds. Importantly, the physics-guided PG-NLNN model maintains

significantly greater stability across the entire prediction horizon.

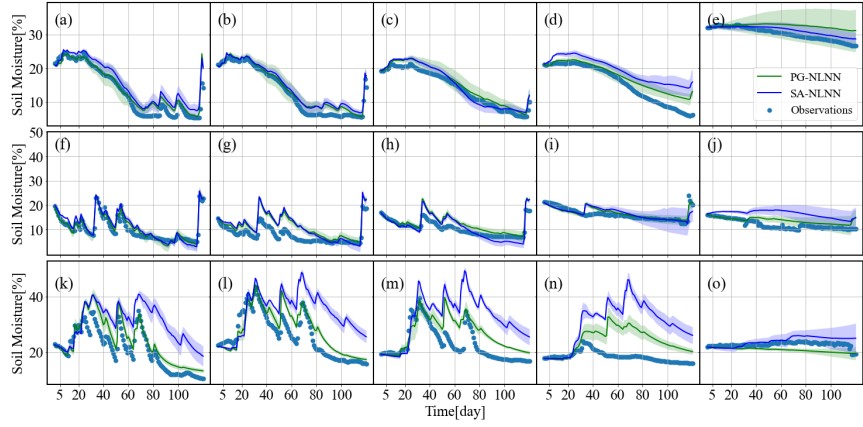

**Figure A2**. The 120-day predicted soil moisture time series of 5 depths with PG-NLNN and SA=NLNN at Falkenberg (a-e), Cape-Charles (f-j), and Goodwell (k-o).




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
