# Peer review of "Interpretable Soil Moisture Prediction with a Physicsguided Deep Learning Approach"

_EGUsphere, 2025_

## Author Comment (AC1)

This manuscript presents an interesting and exploratory study addressing the critical challenge of soil moisture prediction within the hydrological cycle. Specifically, the authors developed two variants of non-local neural networks (NLNN), and the method effectively models vertical heterogeneity and inter-layer connectivity. The authors validated their approach using synthetic data and in-situ observations from the International Soil Moisture Network, highlighting the models' interpretability and their robustness to noise. This approach, which attempts to learn soil water dynamics directly from data without relying on traditional physical assumptions, demonstrates the potential of integrating data-driven techniques with scientific knowledge, particularly in complex soil conditions such as wormholes and root water uptake.

Response:

We are very grateful to receive such valuable comments and suggestions provided by the reviewer regarding our manuscript. We have attempted to address the comments and revised our manuscript.

However, despite the innovative research direction, the manuscript suffers from several issues, particularly in terms of the clarity of the methods. My specific comments are as follows:

1. The claim of a "physics-guided" approach is a core theme, but the method lacks evidence of incorporating actual physical laws into the model structure or loss functions. The model constructed in this study essentially relies on data-driven feature interaction learning. The authors should either replace the term "physics-guided" with a more appropriate expression or explicitly incorporate more concrete physical information into the model.

Response1:

Thank you for your comments. We agree that our model is not driven by explicit physical equations or mechanisms. Instead, its design is guided by fundamental knowledge of soil water dynamics. Therefore, we have replaced the term "physics-guided" with the more accurate descriptor "knowledge-guided" throughout the manuscript to better reflect this approach.

2. The authors mention various deep learning models in the introduction, including CNN, LSTM, Transformer, etc., and state that "the complex coupling of actual physical processes and the presence of unknown governing equations pose substantial challenges in practical applications," based on which they propose the use of NLNNs in this paper. However, Graph Neural Networks (GNNs) can also incorporate spatial relationships into the model. The authors should expand the literature review to more clearly highlight the uniqueness of their approach.

Response2:

Thank you for your comments. We provided a detailed discussion of the differences between graph neural networks and our method NLNNs, highlighting NLNNs' relative advantages:

"As an example, graph neural networks (GNNs) (Scarselli et al., 2008) utilize the adjacency matrix to aggregate node features and achieve local invariance. Wang et al. proposes a spatiotemporal graph convolutional network that models inter-station relationships to effectively predict soil moisture (W. Wang et al., 2025). GNNs rely on explicit, pre-defined graph structures, where neighbor nodes typically share the same transformation rules and the topological relationships remain fixed. In contrast, the Non-local Neural Networks (NLNNs) dynamically compute global dependencies (X.

Wang et al., 2018). Essentially, the non-local operation in NLNNs calculates responses at specific locations by aggregating features from all positions in the input feature map (X. Wang et al., 2018). The inherent design enables NLNNs to flexibly model relationships between variables according to the requirements, functioning like a more flexible GNN on a fully-connected graph. Considering the complexity of interactions between multi-depth soil moisture, we introduce the NLNNs to capture spatially invariant soil moisture relationships across soil layers. Our objective is to model vertical heterogeneity and inter-layer connectivity without physical assumptions."

3. The authors include some conclusive statements in the introduction (e.g., "By integrating meteorological conditions and the spatial interactions of soil moisture within its four-part disentangled physics-guided operation framework, PG-NLNN demonstrates superior performance"). The introduction should mainly address the research background, motivation, scientific problems, and research content. Including such statements is useful for emphasizing the potential contribution of the paper, but it would be more appropriate to place them in the results or conclusion sections.

Response3:

Thank you for your comments. We have revised the introduction by removing these conclusive claims. The relevant content has been moved and integrated into the Discussion and Conclusion sections to provide a more appropriate interpretation of the findings.

4. The time dependency assumption presented in Section 2.1 needs further explanation. The sentence, "In our soil moisture forecasts at multiple depths, we assume that the soil moisture within the profile at the next time step depends on both the current meteorological conditions and the soil moisture from the previous time step," suggests that the soil moisture at time $t1$ and the meteorological conditions at time $t2$ determine the soil moisture at time $t3$. However, it is unclear why the soil moisture at time $t2$ does not influence the soil moisture at time $t3$. Please clarify the time dependency and provide the theoretical foundation for this assumption.

Response4:

Thank you for your comments. To further explain why we design model this way, we have added the physical background and theoretical foundation of the soil water transport equation in the manuscript in section 2.1:

"The dynamics of soil moisture transport are fundamentally described by the Richards equation, a governing relation derived from the mass conservation law and the Buckingham-Darcy law (Buckingham, 1907). For one-dimensional uniform flow in homogeneous soil, and under the assumptions that preferential flow, this equation takes the following form:

$$\frac{\partial \theta}{\partial t} = \frac{\partial}{\partial z}\left[K\left(\frac{\partial \psi}{\partial z} + 1\right)\right] \tag{1}$$

where $\theta\ [cm^3 cm^{-3}]$ is the volumetric moisture content, $t\ [day]$ denotes the time, $z\ [cm]$ is the vertical coordinate (positive upward), $K\ [cm/day]$ is the unsaturated hydraulic conductivity, $\psi\ [cm]$ is the soil matric potential of water.

Based on this equation, the soil moisture profile at a subsequent time step evolves from the preceding profile. Infiltration and evaporation, driven by meteorological factors, directly influence surface soil moisture, which triggers a redistribution of moisture through the soil profile. Therefore, the multi-depth soil moisture at the next time step can be determined by both the current meteorological conditions and the soil moisture profile from the previous time step."

5. The input symbol in Figure 2 should be consistent. The figure uses $Xt$, while the text uses $smt$, which creates an inconsistency. It is recommended to unify the notation for better understanding for readers. Additionally, elements like $Wk$, $Wq$, and $Wv$ in the figure are not fully explained. It would be helpful to expand the caption to provide more detailed descriptions, aiding readers in better understanding the model structure.

Response5:

Thank you for your comments. We have updated Figure 2 by integrating the correspondence between $X^t$ and $sm^t$ to ensure uniformity. Besides, we have added more details about $Wk$, $Wq$, and $Wg$ in the caption as follows:
"

[Figure]

Figure 2. Left: non-local neural network structure for soil moisture forecasting. Right: embedded Gaussian operation and knowledge-guided non-local operation. RPE: relative position encoding. SA/ PG score: non-local weights computed through embedded Gaussian operation and knowledgeguided operation. $W_q$, $W_k$ and $W_g$ are the weight matrixes to be learned for input embeddings.”

6. The model's Physics-guided Operation section utilizes different mask layers to filter out specific data points while emphasizing useful information. This is similar to the self-attention mechanism in Transformers but with domain-specific adjustments. However, the authors have not provided the physical basis for this mechanism, nor have they discussed its advantages compared to the traditional Transformer mechanism. It is recommended to include the theoretical background of this mechanism and explain its specific advantages for soil moisture prediction to enhance the persuasiveness and originality of the work.

Response6:

Thank you for your comments.

The vertical movement of soil water exhibits directional divergence: downward movement is dominated by gravity, as a dissipation of potential energy, while upward movement relies on capillary forces and other mechanisms that work against gravity. On this physical basis, the soil structure (e.g., stratification and cracks) collectively regulate the distribution of moisture. All these moisture interactions are intricately related to the soil moisture content and its vertical depth.

In this specific context, we employ different masks to decouple the soil moisture interactions from different directions. The four masks in Figure 2 are designed for meteorological forcing, upper soil water interactions, same-depth soil moisture interactions, and lower soil water interactions, respectively. Each process is modeled by a fully connected network that takes soil moisture content and depth differences as inputs, ultimately forming the final model. Unlike the self-attention mechanism, this knowledge-guided design separates the significantly different moisture movement processes for independent learning, thereby better capturing the relationships between soil moisture variables. In the future, we will develop more suitable and refined solutions tailored to the specific characteristics of soil water movement.

We will incorporate this paragraph into section 2.2.2 of the manuscript:

“In this work, we propose KG-NLNN, a model specifically designed for forecasting soil moisture at multiple depths in the soil profile, as depicted in Figure 2. The vertical movement of soil moisture exhibits a directional divergence: downward flow is driven primarily by gravity and constitutes a dissipation of potential energy, while upward movement is governed by capillary forces and other mechanisms acting against gravity. In this specific context, we employ a set of masks to decouple soil moisture interactions from different directions. The four masks in Figure 2 correspond to four key components: meteorological forcing, upper soil water interactions, same-depth soil moisture interactions, and lower soil water interactions, respectively. Each of these components is modeled by a fully connected network, which takes soil moisture content and depth differences as inputs. This knowledge-guided architecture separates different moisture movement processes for independent learning, thereby enhancing the model's ability to capture complex relationships among soil moisture variables across the soil profile.”

7. In line 249 and in formulas (7), (8), and (9), $f(xi, xj)$ involves two variables, but the subsequent description mentions that the function is a mapping of three variables. The equation should either be modified or the description clarified to specify the actual number of input variables, ensuring consistency between the equation and the text.

Response7:

Thank you for your comments. We have modified the variables in formulas 7-9 and 12-13:

$$f_0(x_i, x_j, z_i) = FNN_0(x_j, x_i, z_i), j = 0 \tag{8}$$

$$f_u(x_i, x_j, z_i, z_j) = FNN_u(x_i - x_j, x_i, z_i - z_j), i > j \tag{9}$$

$$f_l(x_i, x_j, z_i, z_j) = FNN_l(x_i - x_j, x_i, z_i - z_j), i < j \tag{10}$$

$$f(x_i, x_j, z_i, z_j) = e^{f_0(x_i, x_j, z_i)/N + f_u(x_i, x_j, z_i, z_j)/N + f_l(x_i, x_j, z_i, z_j)/N + f_r(x_i, x_j)/\sqrt{d_k}} \tag{13}$$

$$C(x) = \sum_{\forall j} f(x_i, x_j, z_i, z_j) \tag{14}$$

8. In lines 276–277, the authors mention that the LSTM input contains data from two time steps due to the delayed effect of meteorology on soil moisture. However, the authors do not explain why two time steps were selected or provide any physical or empirical justification. It is recommended to add the rationale and reasoning for this time window choice.

Response8:

Thank you for your comments. The ISMN soil moisture data in this study are derived from 24-hour averages, while precipitation data represent 24-hour cumulative totals. Hydrologically, meteorological conditions from the previous time step (t-1) do not cease their influence immediately at the end of the period. Instead, through processes such as infiltration, lateral flow, and redistribution, they continue to affect soil moisture at the following time step t. Thus, incorporating both time steps enables the model to capture cross-day causal relationships. We set the time step to 2, ensuring that the meteorological input remains both concise and informationally rich. Therefore, the task of learning the meteorological time dependencies is delegated to the LSTM network, which also justifies its application in processing boundary conditions. We have added the relevant content at Line of the manuscript:

"Hydrologically, meteorological conditions from the previous time step (t–1) do not cease their influence immediately; rather, processes such as infiltration, lateral flow, and redistribution allow these conditions to continue affecting soil moisture at the subsequent time step t. Incorporating both time steps thus enables the model to capture cross-day causal relationships. A time step of 2 is used to keep the meteorological inputs concise while retaining adequate informational richness. Accordingly, the task of learning meteorological temporal dependencies is assigned to the LSTM

network, which also justifies its use in processing boundary conditions."

9. Formulas (6) and (14–16) both use the symbol $a$ to represent the activation function, but the authors have not clarified whether the same function is used in both instances. If the activation functions differ, the notation should be distinguished to avoid confusion. Additionally, the explanation of LSTM variables lacks a description of the tanh activation function. It would be beneficial to add this explanation to maintain consistency between the notation and the equation definitions.

Response9:

Thank you for your comments. We have added corresponding subscripts to the activation function symbol "$a$" to differentiate between the tanh and sigmoid activation functions, represented as $a_t$ and $a_s$. Besides, tanh in the equations for LSTM has been revised into $a_t$ to maintain consistency: "The operation of an LSTM can be summarized as follows:

$$i^t = a_s(W_i \cdot [h^{t-1}, ub^t] + b_i) \tag{14}$$

$$f^t = a_s(W_f \cdot [h^{t-1}, ub^t] + b_f) \tag{15}$$

$$o^t = a_s(W_o \cdot [h^{t-1}, ub^t] + b_o) \tag{16}$$

$$\widetilde{C}^t = a_t(W_c \cdot [h^{t-1}, ub^t] + b_c) \tag{17}$$

$$c^t = f^t \cdot c^{t-1} + i^t \cdot \widetilde{C}^t \tag{18}$$

$$h^t = o^t \cdot a_t(c^t) \tag{19}$$

where $W_i$ and $b_i$, $W_f$ and $b_f$, $W_o$ and $b_o$ denote the deep learning parameters for the input gate, forget gate, and the output gate, respectively; $W_c$ and $b_c$ are the parameters for cell state updating; in addition, $i^t$, $f^t$ and $o^t$ are the input gate, forget gate, and output gate at time $t$, respectively, and $c^t$ is the memory cell state; $h^t$ represents the hidden state; $a_s$ is the sigmoid activation function, and $a_t$ denotes the tanh activation function."

10. In Section 3.1, the authors use reference evapotranspiration rather than actual evapotranspiration, which better reflects actual water consumption. It is recommended to clarify the reasoning behind this choice.

Response10:

Thank you for your comments. Actual evapotranspiration (AET) indeed more accurately represents the e soil water consumption. While actual evapotranspiration is intrinsically dependent on soil water content, the creation of virtual scenarios requires an independent and controllable evapotranspiration value, rather than one coupled with dynamic soil moisture conditions. As standardized in the FAO-56 guidelines (Allen et al., 1998), AET is calculated as AET = Kc × PET, where Kc serves as a refined empirical parameter. When generating synthetic data, we applied this empirical coefficient method to derive a preliminary evapotranspiration estimate, adopting a coefficient value of 1.0 in this instance. We have added the corresponding descriptions in Lines:

"For boundary conditions, the daily reference evapotranspiration (ET0) is calculated with the FAO Penman-Monteith method (Allen et al., 1998) in Wuhan coordinates to generate the synthetic data. As standardized in the FAO guidelines (Allen et al., 1998), actual evapotranspiration is the product of $K_C$ and ET0, where $K_C$ serves as a refined empirical parameter. When generating synthetic data, we applied this empirical coefficient method to derive a preliminary evapotranspiration estimate, adopting a coefficient value of 1.0 in this instance."

11. In line 346, the authors list meteorological input variables but do not specify their time and spatial resolutions. It is recommended to provide this information and explain whether the data have been resampled to ensure the completeness and reproducibility of the model input descriptions.
Response11:

Thank you for your comments. The meteorological data we applied in the model input were downloaded at a daily temporal resolution for each specific location based on its latitude and longitude coordinates. We have provided additional clarification in Line:

"The meteorological inputs for our models include precipitation, atmospheric temperature, long-wave radiation, short-wave radiation, wind speed, and relative humidity, as mentioned above. These meteorological data are sourced from the NASA Prediction of Worldwide Energy Resources project (https://power.larc.nasa.gov/). Based on the latitude and longitude coordinates of each station, we downloaded the corresponding point-scale, daily-resolution meteorological datasets."

12. It is recommended to adjust the placement of the legends in Figures 6, 7, 11, and 14, especially in Figures 7 and 14, where it would be better to position the legend at the top of the figure to avoid obscuring important elements of the figure.
Response12:

Thank you for your comments. We have modified the legends of Figures 6, 9, 11, and 14 to present the results more clearly. In accordance with another reviewer's comment, we have replaced Figure 7 with a matrix weight map for enhanced illustration of the model's interpretability in section 4.1. Besides, we have provided comparisons with the LSTM models in Figure 6 and Figure14. The figures with updated legends are presented below.

[Figure]

**Figure 6.** The RMSE results for 1, 3, 7, and 15-day for heterogeneous soil(a-e), and two-layered soil (f-j). The error bar indicates the standard deviations of the RMSE, which are computed via ten training replicates.

[Figure]

**Figure 9.** The average RMSE comparisons between LSTM_4, LSTM_1, SA-NLNN, and KG-NLNN across twenty research sites at 5 depths: 0.05m(a), 0.10m(b), 0.20m(c), 0.50m(d), 1.00m(e).

[Figure]

**Figure 11.** The autoregressive 24-day predicted soil moisture time series of 5 depths with LSTM_1, LSTM_4, KG-NLNN and SA-NLNN at Falkenberg (a-e), Cape-Charles (f-j), and Goodwell (k-o). The shaded region represents the confidence interval of the models, spanning 1 standard deviation.

[Figure]

**Figure 14.** The RMSE results for 1, 3, 7, and 15-day at 0.05m(a-d), 0.10m(e-h), 0.20m(i-l), 0.50m(m-p) and 1.0m(q-t) in the homogenous soil under increasing noise levels through 4 models: SA-NLNN, KG-NLNN, LSTM_1 and LSTM_4. The error bar indicates the standard deviations of the RMSE, which are computed via ten training replicates.

13. In line 409, the term "significant errors" is used, but no statistical support is provided. It is recommended to either include p-values or replace "significant" with terms like "obvious" to avoid using the term without statistical backing.

Response13:

Thank you for your comments. As suggested, we have replaced "significant" with "obvious".

Reference:

Allen, R.G., Pereira, L.S., Raes, D., Smith, M., 1998. Crop evapotranspiration-Guidelines for computing crop water requirements-FAO Irrigation and drainage paper 56. Fao, Rome 300,

D05109.

Buckingham, E., 1907. Studies on the movement of soil moisture.

Veličković, P., Cucurull, G., Casanova, A., Romero, A., Lio, P., Bengio, Y., 2017. Graph attention networks. arXiv Prepr. arXiv1710.10903.

Wang, W., Wei, Y., Hao, L., Wei, Z., Zhao, T., 2025. Soil moisture forecasting in wireless sensor networks via spatiotemporal graph convolutional networks 1–17. https://doi.org/10.1002/vzj2.70000

---

## Author Comment (AC2)

This is a review of the manuscript "Interpretable Soil Moisture Prediction with a Physics-Guided Deep Learning Approach." The authors propose non-local neural networks (NLNNs) for single-time-step, multi-depth soil-moisture forecasts, with two variants: a self-attention NLNN (SA-NLNN) and a physics-guided NLNN (PG-NLNN) that disentangles four influences (upper boundary, upper layers, same-depth memory, lower layers) motivated by gravity, capillarity, and retention. They test their models on both synthetic and field data. They compare the performance of the two models with LSTM baselines and show that the prediction uncertainty is smaller for the proposed NN models than for the LSTM models. They also show that the learned non-local weight matrices can be related to soil texture. This is an interesting direction of research.

I believe this manuscript contains many good ideas for further investigation. Thus, I encourage publication after a major revision.

**Response:**

We sincerely appreciate the reviewer's time and effort in evaluating our manuscript and providing insightful comments. The comments have significantly improved the quality of our work. Since our model is governed not by explicit physical equations but by fundamental knowledge of soil water dynamics, we have replaced the term "physics-guided" with the more accurate descriptor "knowledge-guided" throughout the manuscript to better reflect this approach.

**Major points**

Synthetic vs. field data

It would be more natural to use the same set of models for both synthetic and field data. LSTM baselines are evaluated on field data; the synthetic section compares SA-NLNN vs. PG-NLNN but omits LSTM models on the same synthetic tasks. This weakens causal attribution of PG-NLNN's gains to physics guidance rather than dataset characteristics.

**Response1:**

Thank you for your comments. We have provided the performance of the LSTM models on four synthetic cases and compared them with our proposed models. Table R1 displays the MAE values for 1, 3, 7, and 15-day forecasts of four models. Figure R1 visually displays their RMSE results for both heterogeneous and layered soil scenarios. As shown in Table R1, the LSTM\_4 model achieves very high accuracy in 1-day predictions, but its performance deteriorates rapidly over longer periods. As for the other models, NLNNs and LSTM\_1 exhibit comparable performance, with LSTM\_1 in fact outperforms them in certain scenarios, for example, in the case of root water uptake. To further investigate and compare these four models, we evaluate them under five levels of noise (0.5, 1.0, 2.0, 5.0, 10.0). The RMSE results for soil moisture prediction at 0.05m, 0.10m, 0.20m, 0.50m, and 1.00m are presented in Figure R2. The LSTM\_4 model demonstrates poor noise resistance and long-term forecasting capability. The other three models perform similarly under low-noise conditions, with LSTM\_1 even exhibiting some advantage. However, as the noise level increases, the two NLNN models demonstrate better robustness. Notably, the knowledge-guided NLNN is particularly stable, consistent with its performance on in-situ soil moisture data.

Tabel R1. The MAE [%] values for 1, 3, 7, and 15-day forecasts of LSTM\_4, LSTM\_1, the proposed KG-NLNN and SA-NLNN model at 5 depths under four designed scenarios.

| Depth/m |                           |        |        |       |               |       |       | KG-N  | ILNN        |       |       |       |                   |       |       |       |
|---------|---------------------------|--------|--------|-------|---------------|-------|-------|-------|-------------|-------|-------|-------|-------------------|-------|-------|-------|
|         | homogeneous heterogeneous |        |        |       |               |       |       |       | two-lavered |       |       |       | root water uptake |       |       |       |
|         | 1d                        | 3d     | 7d     | 15d   | 1d            | 3d    | 7d    | 15d   | 1d          | 3d    | 7d    | 15d   | 1d                | 3d    | 7d    | 15d   |
| 0.05    | 0.235                     | 0.327  | 0.433  | 0.539 | 0.259         | 0.372 | 0.510 | 0.652 | 0.449       | 0.680 | 0.945 | 1.170 | 0.528             | 0.747 | 0.996 | 1.224 |
| 0.10    | 0.313                     | 0.451  | 0.627  | 0.788 | 0.306         | 0.431 | 0.593 | 0.749 | 0.521       | 0.745 | 0.995 | 1.191 | 0.409             | 0.544 | 0.685 | 0.825 |
| 0.20    | 0.342                     | 0.533  | 0.776  | 1.016 | 0.305         | 0.488 | 0.736 | 0.971 | 0.433       | 0.649 | 0.901 | 1.179 | 0.623             | 0.852 | 1.056 | 1.212 |
| 0.50    | 0.235                     | 0.357  | 0.545  | 0.782 | 0.253         | 0.399 | 0.630 | 0.952 | 0.334       | 0.518 | 0.774 | 1.098 | 0.375             | 0.598 | 0.870 | 1.182 |
| 1.00    | 0.203                     | 0.312  | 0.445  | 0.647 | 0.244         | 0.397 | 0.618 | 0.934 | 0.368       | 0.625 | 0.969 | 1.329 | 0.278             | 0.455 | 0.749 | 1.120 |
| Depth/m | SA-NLNN                   |        |        |       |               |       |       |       |             |       |       |       |                   |       |       |       |
|         |                           | homog  | eneous |       | heterogeneous |       |       |       | two-layered |       |       |       | root water uptake |       |       |       |
|         | 1d                        | 3d     | 7d     | 15d   | 1d            | 3d    | 7d    | 15d   | 1d          | 3d    | 7d    | 15d   | 1 d        | 3d    | 7d    | 15d   |
| 0.05    | 0.328                     | 0.470  | 0.686  | 1.039 | 0.363         | 0.524 | 0.750 | 1.840 | 0.327       | 0.505 | 0.836 | 2.210 | 0.536             | 0.918 | 2.150 | 6.702 |
| 0.10    | 0.249                     | 0.375  | 0.580  | 0.957 | 0.220         | 0.314 | 0.477 | 0.851 | 0.390       | 0.569 | 0.808 | 1.465 | 0.322             | 0.447 | 0.675 | 1.480 |
| 0.20    | 0.262                     | 0.366  | 0.519  | 0.820 | 0.292         | 0.389 | 0.482 | 0.648 | 0.487       | 0.696 | 0.945 | 1.350 | 0.379             | 0.546 | 0.775 | 1.861 |
| 0.50    | 0.209                     | 0.291  | 0.414  | 0.566 | 0.265         | 0.337 | 0.431 | 0.623 | 0.327       | 0.483 | 0.708 | 1.018 | 0.344             | 0.485 | 0.687 | 1.502 |
| 1.00    | 0.245                     | 0.376  | 0.575  | 0.807 | 0.282         | 0.430 | 0.640 | 0.941 | 0.336       | 0.530 | 0.810 | 1.250 | 0.297             | 0.482 | 0.820 | 1.748 |
| Depth/m |                           | LSTM 4 |        |       |               |       |       |       |             |       |       |       |                   |       |       |       |
|         | homogeneous               |        |        |       | heterogeneous |       |       |       | two-layered |       |       |       | root water uptake |       |       |       |
|         | 1d                        | 3d     | 7d     | 15d   | 1d            | 3d    | 7d    | 15d   | 1d          | 3d    | 7d    | 15d   | 1d                | 3d    | 7d    | 15d   |
| 0.05    | 0.009                     | 1.503  | 2.791  | 3.874 | 0.010         | 1.520 | 2.829 | 3.936 | 0.015       | 1.422 | 2.795 | 4.106 | 0.020             | 1.649 | 3.187 | 4.641 |
| 0.10    | 0.007                     | 1.176  | 2.237  | 3.184 | 0.007         | 1.202 | 2.282 | 3.240 | 0.015       | 1.479 | 2.879 | 4.179 | 0.014             | 1.255 | 2.449 | 3.584 |
| 0.20    | 0.008                     | 0.786  | 1.630  | 2.380 | 0.010         | 0.782 | 1.628 | 2.384 | 0.012       | 0.836 | 1.735 | 2.561 | 0.013             | 0.801 | 1.671 | 2.479 |
| 0.50    | 0.006                     | 0.406  | 0.942  | 1.483 | 0.008         | 0.373 | 0.872 | 1.375 | 0.008       | 0.400 | 0.933 | 1.476 | 0.009             | 0.403 | 0.939 | 1.482 |
| 1.00    | 0.008                     | 0.266  | 0.662  | 1.116 | 0.007         | 0.266 | 0.664 | 1.121 | 0.006       | 0.258 | 0.644 | 1.103 | 0.006             | 0.267 | 0.667 | 1.136 |
| Depth/m |                           |        |        |       |               |       |       | LST   | M 1         |       |       |       |                   |       |       |       |
|         | homogeneous               |        |        |       | heterogeneous |       |       |       | two-layered |       |       |       | root water uptake |       |       |       |
|         | 1d                        | 3d     | 7d     | 15d   | 1d            | 3d    | 7d    | 15d   | 1d          | 3d    | 7d    | 15d   | 1d                | 3d    | 7d    | 15d   |
| 0.05    | 0.318                     | 0.440  | 0.590  | 0.845 | 0.346         | 0.484 | 0.656 | 0.948 | 0.264       | 0.451 | 0.771 | 1.313 | 0.343             | 0.462 | 0.600 | 0.804 |
| 0.10    | 0.135                     | 0.202  | 0.319  | 0.528 | 0.149         | 0.249 | 0.408 | 0.699 | 0.359       | 0.542 | 0.863 | 1.436 | 0.274             | 0.365 | 0.491 | 0.707 |
| 0.20    | 0.120                     | 0.174  | 0.262  | 0.444 | 0.138         | 0.217 | 0.359 | 0.669 | 0.218       | 0.320 | 0.545 | 1.072 | 0.159             | 0.238 | 0.366 | 0.594 |
| 0.50    | 0.128                     | 0.177  | 0.274  | 0.494 | 0.142         | 0.207 | 0.341 | 0.640 | 0.179       | 0.293 | 0.542 | 1.090 | 0.173             | 0.276 | 0.443 | 0.742 |
| 1.00    | 0.214                     | 0.350  | 0.594  | 1.075 | 0.188         | 0.288 | 0.465 | 0.865 | 0.180       | 0.293 | 0.578 | 1.343 | 0.242             | 0.393 | 0.672 | 1.203 |

Figure R1. The RMSE results for 1, 3, 7, and 15-day for heterogeneous soil(a-e), and two-layered soil (f-j). The error bar indicates the standard deviations of the RMSE, which are computed via ten training replicates.

Figure R2. The RMSE results for 1, 3, 7, and 15-day at 0.05m(a-d), 0.10m(e-h), 0.20m(i-l), 0.50m(m-p) and 1.00m(q-t) for homogenous soil under increasing noise levels of the LSTM\_4, LSTM\_1, SA-NLNN and KG-NLNN models. The error bar indicates the standard deviations of the RMSE, which are computed via ten training replicates.

To improve readability and focus on model interpretability in section 4.1-- given that the extensive statistical data in Table R1 could be overwhelming for synthetic data analysis-- we have moved Table R1 to the appendix. Therefore, Figure R1 now represents the models' comparative accuracy and replaces the original Figure 6. The original Figure 7 has been removed. Furthermore, Figure 14 in the manuscript has been replaced by Figure R2 to provide a more comprehensive evaluation of noise resistance, including a direct comparison with LSTM models.

**Interpretability not yet quantitatively tied to physics**

Weight maps qualitatively reflect layering, but the link to soil texture/parameters is not quantified (e.g., correlation with (Ksat) contrasts or van Genuchten parameters across cases/sites). Response2:

**Thank you for your comments.**

As we mentioned in introduction, our models are not based on any parameterizations and assumptions, instead attempt to learn the soil water dynamics directly from the data. Therefore, it is difficult to yield quantitative descriptions of van Genuchten parameters from the weight maps. Soils

with different van Genuchten parameters (such as Ks) will also exhibit distinct patterns in the weight map. We will provide the following two examples to demonstrate that differences in soil parameters are reflected in the weight map:

Figure R3 (Figure 8 in the manuscript) depicts the weight matrix maps generated by the KG-NLNN model for two-layered soil scenarios. This case is created by interchanging the properties of the upper and lower soil layers, with Ks values set at 0.25 and 10.49, respectively, and the rest of the parameters identical, as recorded in Appendix A. The saturated hydraulic conductivity of the two soil types varies significantly, with distinct characteristics influencing water transport and drainage. With sand below loam, water from the upper loam percolates quickly and deeply, making lower-layer moisture dominated by the upper loam. As shown in Figure R3 (a), the moisture in the lower layer (0.10m, 0.20m, 0.5m, 1.0m) is notably influenced by the moisture at 0.05m. Conversely, with sand above loam, the upper sand rapidly drains water, and the water from the upper sand is absorbed and held by the lower loam. Therefore, soil moisture in the lower layer is mainly affected by the adjacent upper layer, as shown in Figure R3(b). This layered pattern, evident in the weight map, serves as a qualitative indicator of soil texture. Although this cannot provide a quantitative description of the soil hydraulic parameters, it can reflect the difference in hydraulic conductivity between the layers and reveal which layer is more permeable.

Figure R3. The non-local weight maps in two-layered simulated stratified soil scenarios through KG-NLNN (a) loam above sand (b) sand above loam.

Additionally, we have included a new case in the synthetic data that compares the weight maps of homogeneous soils with different Ks values. The corresponding non-local weight maps derived from KG-NLNN are shown in Figure R4. Differing hydraulic conductivities governs soil water flow velocity, which causes variations in the time required for water to reach various depths This shapes the formation of the weight maps, leading to the distinctive patterns observed in Figure R4(a) and (b). For instance, loam (Ks = 0.25) exhibits slow infiltration, so its moisture content is easily influenced by adjacent layers in Figure R4(a). In contrast, sand (Ks=10.49) allows rapid infiltration, resulting in deeper soil moisture being affected directly by meteorological factors. Thus, although our model does not involve any parameterization nor perform a quantitative description of soil hydraulic parameters, it nevertheless provides insights into these hydraulic properties to some extent.

Figure R4. The non-local weight maps in homogeneous simulated soil scenarios through KG-NLNN (a) Ks = 0.25 (b) Ks = 10.49.

Since our model is not based on any parameterization, our objective is not to quantitatively describe parameters like Ks. Rather, it is to use figures to reflect differences between soils and represent their true states. We have revised the section 4.1 interpretability analysis according to the above.

**Minor points**

Line 159: 'sm 1 ... sm n' are not explicitly defined.

Response3:

Thank you for comments. Here we have provided a clearer definition of  $sm_n^t$  in the manuscript: "Here,  $sm_n^t$  denotes the soil moisture at depth n and time t."

Section 2: How did the authors determine the initial soil moisture for the synthetic and field data cases? Please specify exactly how the first step of multi-day forecasts is initialized in both the synthetic and field experiments

Response4:

Thank you for comments.

**Initial Conditions and Data Generation**

For the synthetic data generating, we set the initial soil moisture content within the top 1 meter to 0.1 to simplify the conditions. The dataset is then evolved under prescribed meteorological conditions with a free-drainage lower boundary condition. We have revised the sentence here:

"The lower boundary condition is set as free drainage, and the initial moisture content of the soil column is set to a uniform value of 0.10."

In the case of real-world scenarios, the initial soil moisture value is taken as the 24-hour average measured at the site on the starting date of the selected time period

**Initialization for Prediction**

For model prediction, the models' initial moisture content input is the truth soil moisture value from the day before the prediction. In the synthetic scenario, this value comes from the physical model's output; in the real-world scenario, it is taken from field observations. We have added the following description in the manuscript:

"The initial soil moisture content for the prediction is set to the truth from the preceding day. Specifically, this value is obtained from the physical model's output for the virtual scenario and from field observations for the real-world scenario."